# Fusion pore regulation by cAMP/Epac2 controls cargo release during insulin exocytosis

Alenka Guček[1], Nikhil R Gandasi[1], Muhmmad Omar-Hmeadi[1], Marit Bakke[2], Stein O Døskeland[2], Anders Tengholm[1], Sebastian Barg[1]*

[1]Department of Medical Cell Biology, Uppsala University, Uppsala, Sweden; [2]Department of Biomedicine, University of Bergen, Bergen, Norway

**Abstract** Regulated exocytosis establishes a narrow fusion pore as initial aqueous connection to the extracellular space, through which small transmitter molecules such as ATP can exit. Co-release of polypeptides and hormones like insulin requires further expansion of the pore. There is evidence that pore expansion is regulated and can fail in diabetes and neurodegenerative disease. Here, we report that the cAMP-sensor Epac2 (Rap-GEF4) controls fusion pore behavior by acutely recruiting two pore-restricting proteins, amisyn and dynamin-1, to the exocytosis site in insulin-secreting beta-cells. cAMP elevation restricts and slows fusion pore expansion and peptide release, but not when Epac2 is inactivated pharmacologically or in Epac2$^{-/-}$ (*Rapgef4*$^{-/-}$) mice. Consistently, overexpression of Epac2 impedes pore expansion. Widely used antidiabetic drugs (GLP-1 receptor agonists and sulfonylureas) activate this pathway and thereby paradoxically restrict hormone release. We conclude that Epac2/cAMP controls fusion pore expansion and thus the balance of hormone and transmitter release during insulin granule exocytosis.
DOI: https://doi.org/10.7554/eLife.41711.001

*For correspondence:
sebastian.barg@mcb.uu.se

Competing interests: The authors declare that no competing interests exist.

## Introduction

Insulin is secreted from pancreatic β-cells and acts on target tissues such as muscle and liver to regulate blood glucose. Secretion of insulin occurs by regulated exocytosis, whereby secretory granules containing the hormone and other bioactive peptides and small molecules fuse with the plasma membrane. The first aqueous contact between granule lumen and the extracellular space is a narrow fusion pore (upper limit 3 nm; *Albillos et al., 1997*) that is thought to consist of both lipids and proteins (*Bao et al., 2016*; *Sharma and Lindau, 2018*). At this stage, the pore acts as a molecular sieve that allows release of small transmitter molecules such as nucleotides and catecholamines, but traps larger cargo (*Obermüller et al., 2005*; *Barg et al., 2002*; *MacDonald et al., 2006*; *Alvarez de Toledo et al., 1993*). Electrophysiological experiments have shown that the fusion pore is short-lived and flickers between closed and open states, suggesting that mechanisms exist that stabilize this channel-like structure and restrict pore expansion (*MacDonald et al., 2006*; *Hanna et al., 2009*; *Breckenridge and Almers, 1987*; *Lollike et al., 1995*). The pore can then expand irreversibly (termed full fusion), which leads to mixing of granule and plasma membrane and release of the bulkier hormone content (*Obermüller et al., 2005*; *Barg et al., 2002*; *Anantharam et al., 2010*). Alternatively, the pore can close indefinitely to allow the granule to be retrieved, apparently intact, into the cell interior (termed kiss-and-run or cavicapture) (*Obermüller et al., 2005*; *MacDonald et al., 2006*; *Taraska et al., 2003*; *Tsuboi and Rutter, 2003*; *Shin et al., 2018*). Estimates in β-cells suggest that 20–50% of all exocytosis in β-cells are transient kiss-and-run events that do not lead to insulin release (*Obermüller et al., 2005*; *MacDonald et al., 2006*). However, kiss-and-run exocytosis contributes to local signaling within the islet because smaller granule constituents, such as

**eLife digest** Insulin is the hormone that signals to the body to take up sugar from the blood. Specialized cells in the pancreas – known as β-cells – release insulin after a meal. Before that, insulin molecules are stored in tiny granules inside the β-cells; these granules must fuse with the cells' surface membranes to release their contents. The first step in this process creates a narrow pore that allows small molecules, but not the larger insulin molecules, to seep out. The pore then widens to release the insulin. Since the small molecules are known to act locally in the pancreas, it is possible that this "molecular sieve" is biologically important. Yet it is not clear how the pore widens.

One of the problems for people with type 2 diabetes is that they release less insulin into the bloodstream. Two kinds of drugs used to treat these patients work by stimulating β-cells to release their insulin. One way to achieve this is by raising the levels of a small molecule called cAMP, which is well known to help prepare insulin granules for release. The cAMP molecule also seems to slow the widening of the pore, and Gucek et al. have now investigated how this happens at a molecular level.

By observing individual granules of human β-cells using a special microscope, Gucek et al. could watch how different drugs affect pore widening and content release. They also saw that cAMP activated a protein called Epac2, which then recruited two other proteins – amisyn and dynamin – to the small pores. These two proteins together then closed the pore, rather than expanding it to let insulin out. Type 2 diabetes patients sometimes have high levels of amisyn in their β-cells, which could explain why they do not release enough insulin. The microscopy experiments also revealed that two common anti-diabetic drugs activate Epac2 and prevent the pores from widening, thereby counteracting their positive effect on insulin release. The combined effect is likely a shift in the balance between insulin and the locally acting small molecules.

These findings suggest that two common anti-diabetic drugs activate a common mechanism that may lead to unexpected outcomes, possibly even reducing how much insulin the β-cells can release. Future studies in mice and humans will have to investigate these effects in whole organisms.

DOI: https://doi.org/10.7554/eLife.41711.002

nucleotides, glutamate or GABA, are released even when the fusion pore does not expand. Within the islet, ATP synchronizes β-cells (*Hellman et al., 2004*), and has both inhibitory (*Salehi et al., 2005*; *Poulsen et al., 1999*) and stimulatory (*Richards-Williams et al., 2008*) effects on insulin secretion. ATP suppresses glucagon release from α-cells (*Tudurí et al., 2008*), and activates macrophages (*Weitz et al., 2018*). Interstitial GABA leads to tonic GABA-A receptor activation and α-cell proliferation (*Jin et al., 2013*; *Ben-Othman et al., 2017*), and glutamate stimulates glucagon secretion (*Cabrera et al., 2008*).

Regulation of fusion pore behavior is not understood mechanistically, but several cellular signaling events affect both lifetime and flicker behavior. Pore behavior has been shown to be regulated by cytosolic $Ca^{2+}$, cAMP, PI(4,5)P$_2$, and activation of protein kinase C (PKC) (*MacDonald et al., 2006*; *Hanna et al., 2009*; *Alés et al., 1999*; *Calejo et al., 2013*; *Scepek et al., 1998*) and recent superresolution imaging indicates that elevated $Ca^{2+}$ and dynamin promote pore closure (*Shin et al., 2018*; *Chiang et al., 2014*). Both myosin and the small GTPase dynamin are involved in fusion pore restriction (*Jackson et al., 2015*; *Tsuboi et al., 2004*; *Graham et al., 2002*; *Artalejo et al., 1995*; *Aoki et al., 2010*), and assembly of filamentous actin promotes fusion pore expansion (*Wen et al., 2016*), suggesting a link to endocytosis and the cytoskeleton. In β-cells of type-2 diabetics, upregulation of amysin leads to decreased insulin secretion because fusion pore expansion is impaired (*Collins et al., 2016*), and the Parkinson's related protein α-synuclein promotes fusion pore dilation in chromaffin cells and neurons (*Logan et al., 2017*), thus providing evidence for altered fusion pore behavior in human disease.

Inadequate insulin secretion in type-2 diabetes (T2D) is treated clinically by two main strategies. First, sulfonylureas (e.g. tolbutamide and glibenclamide) close the K$_{ATP}$ channel by binding to its regulatory subunit SUR1, which leads to increased electrical activity and $Ca^{2+}$-influx that triggers insulin secretion (*Henquin, 2000*). Sulfonylureas are given orally and are first-line treatment for type-2 diabetes in many countries. Second, activation of the receptor for the incretin hormone glucagon-like

peptide 1 (GLP-1) raises cytosolic [cAMP] and thereby increases the propensity of insulin granules to undergo exocytosis. Both peptide agonists of the GLP-1 receptor (e.g. exendin-4) and inhibitors of DPP-4 are used clinically for this purpose. The effect of cAMP on exocytosis is mediated by a protein-kinase A (PKA)-dependent pathway, and by Epac2, a guanine nucleotide exchange factor for the Ras-like small GTPase Rap (*Kawasaki et al., 1998*) that is a direct target for cAMP (*Ozaki et al., 2000*) and is recruited to insulin granule docking sites (*Alenkvist et al., 2017*). Epac2 has also been suggested to be activated by sulfonylureas (*Zhang et al., 2009*), which may underlie some of their effects on insulin secretion.

In the following, we will report two parameters that reflect fusion pore behavior, the fraction of exocytosis events with flash phenotype (indicating restricted pores, about 40% in control conditions; *Figure 1d*), and the duration of the flash, referred to as 'NPY release times'. The latter was estimated by fitting a discontinuous function to the fluorescence timecourse (see *Figure 1c*, green lines and *Figure 1e*), which limits the analysis to granules that eventually released their peptide content. The distribution of the NPY release times followed a mono-exponential function and was on average $0.87 \pm 0.12$ s (186 granules in 26 cells) in control conditions (*Figure 1e*). Such events are increased by elevated cAMP (*MacDonald et al., 2006*; *Hanna et al., 2009*) and likely other conditions that stabilize the fusion pore. Indeed, when forskolin (2 µM; fsk) was added to the bath solution we observed a twofold increase of exocytosis rate (*Figure 1f*), a threefold increase of NPY release times (*Figure 1e*), and a nearly doubled fraction of events with restricted fusion pores (*Figure 1d,f*). The GLP-1 agonist exendin-4 (10 nM; Ex4) had comparable effects (*Figure 1d–f*). Effects similar to those observed for human β-cells (*Figure 1*) were observed in the insulin secreting cell line INS-1 (*Figure 1—figure supplement 1*).

The effect of fsk on fusion pore behavior was mimicked by the specific Epac2 agonist S223 (*Schwede et al., 2015*). Incubation with S223-acetomethoxyester (5 µM) increased the fraction of flash events by 60% (*Figure 1d*), doubled average NPY release times (*Figure 1e*) and doubled the event frequency (*Figure 1f*); the effects of fsk and S223 were not additive. In contrast, the Epac-inhibitor ESI-09 decreased the exocytosis rate in the presence of fsk by 80% (*Figure 1f*), and the average NPY release time and the fraction of flash events were both reduced by 60% (*Figure 1d–e*).

Here, we have studied fusion pore regulation in pancreatic β-cells, using high-resolution live-cell imaging. We report that activation of Epac2, either through GLP1-R/cAMP signaling or via sulfonylurea, restricts expansion of the insulin granule fusion pore by recruiting dynamin and amisyn to the exocytosis site. Activation of this pathway by two classes of antidiabetic drugs therefore hinders full fusion and insulin release, which is expected to reduce their effectiveness as insulin secretagogues.

## Results

### cAMP-dependent fusion pore restriction is regulated by Epac but not PKA

To monitor single granule exocytosis, human pancreatic β-cells were infected with adenovirus encoding the granule marker NPY-Venus and imaged by TIRF microscopy. Exocytosis was evoked by local application of a solution containing 75 mM K⁺, which leads to rapid depolarization and Ca²⁺ influx. Visually, two phenotypes of granule exocytosis were observed. In the first, termed full fusion, fluorescence of a granule that was stably situated at the plasma membrane suddenly vanished during the stimulation (in most cases within <100 ms; *Figure 1a–c*, left panels). Since the EGFP label is relatively large (3.7 nm vs 3 nm for insulin monomers), this is interpreted as rapid pore widening that allowed general release of granule cargo. The sudden release of material may suggest that this release coincided with the collapse of the granule into the plasma membrane, but we cannot exclude that at least some granules remained intact (*Taraska et al., 2003*; *Shin et al., 2018*; *Tsuboi et al., 2004*; *Huang et al., 2018*). In the second type, the rapid loss of the granule marker was preceded by an increase in its fluorescence that could last for several seconds (flash events, *Figure 1a–c*, right panels). We and others have previously shown (*Taraska et al., 2003*; *Ferraro et al., 2005*; *Gandasi and Barg, 2014*) that this reflects neutralization of the acidic granule lumen and dequenching of the EGFP-label, before the labeled cargo is released. Since this neutralization occurs as the result of proton flux through the fusion pore, the fluorescence timecourse of these events can be used to quantitatively study fusion pore behavior.

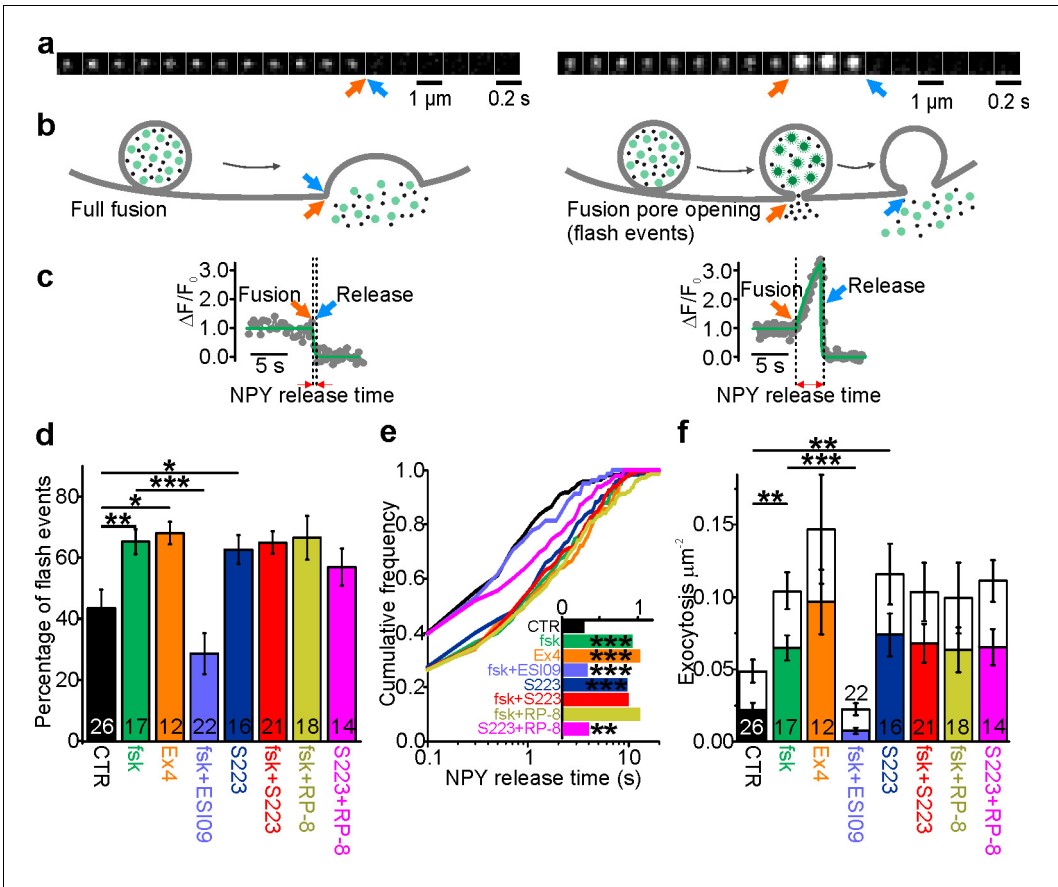

**Figure 1.** cAMP-dependent fusion pore restriction depends on Epac (but not PKA). (**a**) Examples of single granule exocytosis in human β-cells expressing NPY-Venus and challenged with 75 mM K$^+$. Full fusion (left) and flash event (right), where sudden loss of the granule label was preceded by a transient fluorescence increase. Arrows indicate moment of fusion pore opening (orange) and content release (blue). (**b**) Cartoons illustrating the interpretation of events in a. (**c**) Fluorescence time courses for the events in b. Overlaid (green) are fitted functions used to estimate NPY release time. (**d**) Fraction of flash events in experiments as in a-c, in cells exposed to the indicated agents; forskolin (fsk, p=0.01 vs ctrl), exendin-4 (Ex4, p=0.02 vs ctrl), ESI-09 (p=3*10$^{-4}$ vs fsk), S223 (p=0.04 vs ctrl), fsk +S223 (p=0.99 vs fsk), RP-8 (p=0.91 vs fsk) and Rp-8 +S223 (p=0.19 vs ctrl; Kruskal Wallis/Dunn). Number of donors analyzed: 7 (CTR); 5 (fsk); 4 (Ex4); 7 (fsk +ESI09); 6 (S223); 6 (fsk +S223); 7 (RP-8); 4 (Rp-8 +S223). n, number of cells. (**e**) Cumulative frequency histograms of NPY release times; fsk (p=9*10$^{-7}$ vs ctrl), Ex4 (p=1*10$^{-6}$ vs ctrl), ESI-09 (p=2*10$^{-4}$ vs fsk), S223 (p=4*10$^{-6}$ vs ctrl),Fsk +S223 (n.s. vs fsk), RP-8 (n.s. vs fsk) and RP-8 +S223 (p=0.016 vs ctrl; Kolmogorov-Smirnov test). Inset shows median NPY release times for 170 (CTR), 197 (fsk), 155 (Ex4), 81 (ESI-09), 240 (S223), 328 (fsk +S223), 277 (RP-8) and 227 (Rp-8 +S223) events. (**f**) Exocytosis during 40 s of K$^+$-stimulation for control (CTR) and with forskolin (fsk, 2 µM, p=0.002 vs ctrl; Kruskal Wallis/Dunn) or Exendin-4 (Ex4, 10 nM, p=0.005 vs ctrl) or S223 (5 µM, p=0.002 vs ctrl) or RP-8 +S223 (p=0.012 vs ctrl and n.s. vs S223) in the bath solution. Inhibitors of Epac (ESI-09, 10 µM, p=9*10$^7$ vs fsk) or PKA (RP-8, 100 µM, n.s. vs fsk) or Epac2 activator S223 (n.s. vs fsk), one-way ANOVA with Games-Howell post hoc test) were supplied in addition to forskolin. Flash exocytosis (in color) and full fusions (in white) are shown separately. n, number of cells.

DOI: https://doi.org/10.7554/eLife.41711.003

The following figure supplement is available for figure 1:

**Figure supplement 1.** cAMP increases NPY release times in INS1 cells.
DOI: https://doi.org/10.7554/eLife.41711.004

PKA inhibition with Rp8-Br-cAMPS (*Gjertsen et al., 1995*) decreased neither the fraction of flash events, nor average NPY release times (*Figure 1e*). The results indicate that Epac rather than PKA is responsible for cAMP-dependent fusion pore regulation. Paradoxically, Epac activation increases the rate of exocytosis but slows the rate of peptide release from individual granules.

## Epac2 overexpression restricts fusion pores and prolongs their lifetime

We studied the effect of Epac2 overexpression on fusion pore regulation. INS-1 cells were co-transfected with EGFP-Epac2 and NPY-tdmOrange2 and fluorescence was recorded simultaneously in both color channels. Epac2 overexpression had no effect on the overall exocytosis rate in either absence or presence of fsk (*Figure 2a*), but increased the rate of flash events (*Figure 2b–c*), supporting our finding, based on manipulation of the endogenous Epac2 activity, that Epac2 is involved in fusion pore regulation (*Figure 1d*). NPY release times in cells overexpressing Epac2 increased threefold in the absence of fsk, and were similar to controls in presence of fsk (*Figure 2d*). This indicates that a high Epac concentration can achieve sufficient activity to affect insulin secretion even at basal cAMP level, likely because cAMP acts in part by increasing the Epac concentration at the plasma membrane (*Alenkvist et al., 2017*).

## ATP release is accelerated upon Epac inhibition

To test if cAMP-dependent fusion pore restriction affects release of small transmitter molecules, we quantified nucleotide release kinetics from individual granules using patch clamp electrophysiology. The purinergic receptor cation channel $P2X_2$, tagged with RFP ($P2X_2$-RFP), was expressed in INS-1 cells as an autaptic nucleotide sensor (*Obermüller et al., 2005*) (*Figure 3a*). The cells were voltage-clamped in whole-cell mode and exocytosis was elicited by including a solution with elevated free $Ca^{2+}$ (calculated 600 nM) in the patch electrode. In this configuration, every exocytosis event that

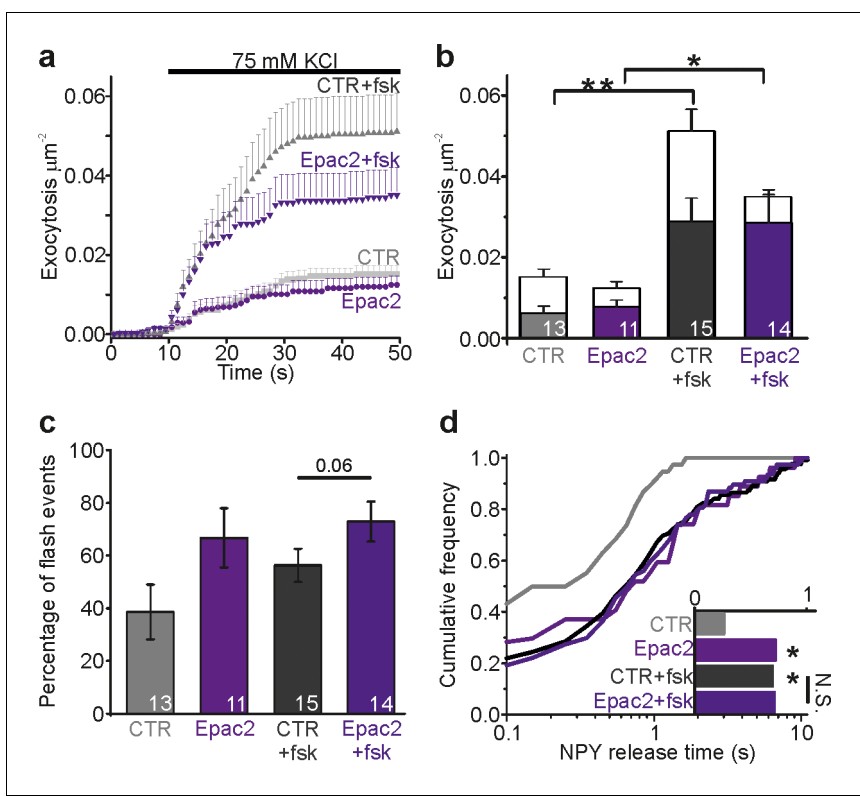

**Figure 2.** Epac2 overexpression prolongs NPY release times. (a) Cumulative exocytosis in INS-1 cells stimulated with 75 mM $K^+$; gray for control cells, purple for cells expressing Epac2-EGFP (both also expressed NPY-tdmOrange2); fsk indicates forskolin in the bath solution. CTR, n = 13 (4 preps); Epac2, n = 11 (2 preps); CTR +fsk, n = 15 (5 preps); Epac2 +fsk, n = 14 cells (2 preps). (b) Total exocytosis in (a), separated into flash events (color) and full fusion (white). Epac2 expression reduced full fusion events (no fsk p=0.06; with fsk p=0.01, Kruskal Wallis/Dunn). n, number of cells. (c) Fraction of flash events in (a–b). (Kruskal Wallis/Dunn). n, number of cells. (d) NPY release times for conditions in a-c. Epac overexpression increased NPY release times in absence (p=0.014) but not in presence of fsk (p=0.87, Kolmogorov-Smirnov test). Inset shows the NPY release times for 38 (CTR), 27 (Epac2), 119 (CTR +fsk) and 77 (Epac2 +fsk) events.

DOI: https://doi.org/10.7554/eLife.41711.005

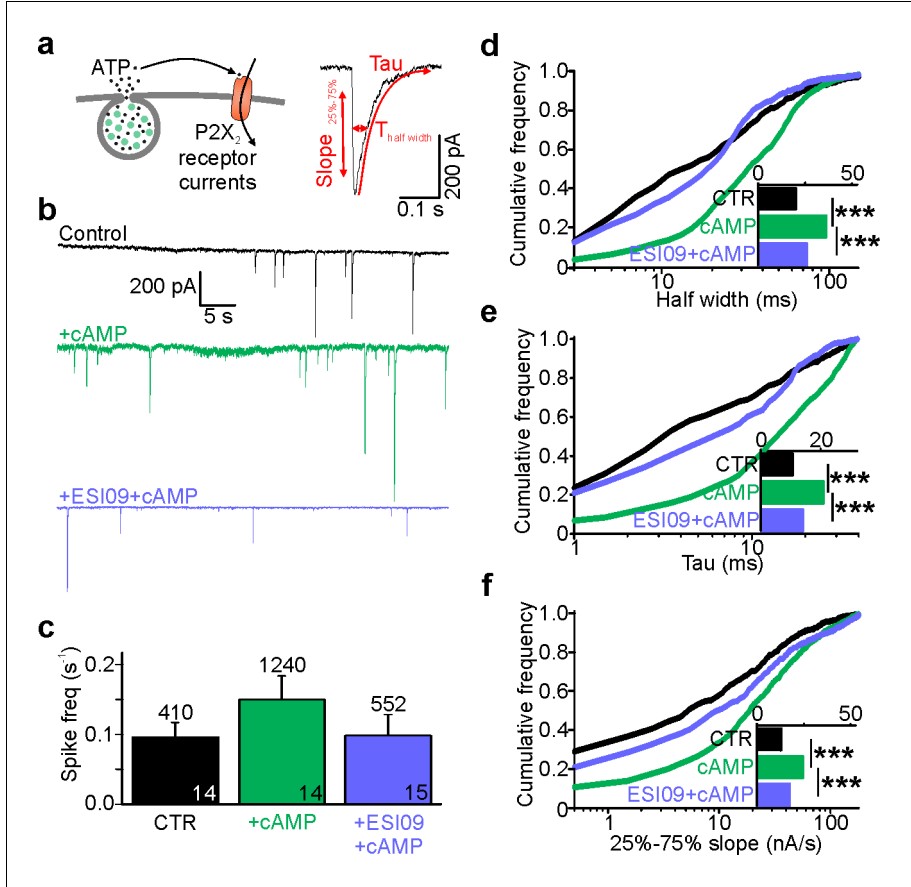

**Figure 3.** Cytosolic cAMP slows ATP release by activating Epac. (a) Electrophysiological detection of nucleotide release events in INS-1 cells expressing P2X$_2$-RFP. Cartoon of the assay (left) and example current spike (black) with fit and analysis parameters (red; T$_{half}$, tau and slope during 25% to 75% of peak). (b) Representative P2X$_2$ currents for control (black), and with cAMP (green) or with cAMP together with ESI-09 (purple) in the electrode solution. (c) Spike frequency conditions in (b). n of events (on top) and n of cells (on bars); two preps for each condition. (d–f) Cumulative frequency histograms of spike half width (d), decay constant tau (e), and slope of the rising phase (25% and 75% of peak, (f)) for CTR (n = 410 spikes, 14 cells), +cAMP (n = 1240, 14 cells) and +ESI-09 + cAMP (n = 552, 15 cells) with medians in the insets. cAMP increased half-width (p=4.1*10$^{-31}$ vs ctrl, Kolmogorov-Smirnov test), tau (p=2.7*10$^{-32}$, Kolmogorov-Smirnov test), and rising slope (p=4.7*10$^{-19}$, Kolmogorov-Smirnov test); the effects were reversed by ESI-09 (p=3.4*10$^{-21}$, p=3.6*10$^{-22}$, and p=1.3*10$^{-9}$, Kolmogorov-Smirnov test), respectively.

DOI: https://doi.org/10.7554/eLife.41711.006

co-releases nucleotides causes an inward current spike, similar to those observed by carbon fiber amperometry (*Figure 3a–b*). Including cAMP in the pipette solution increased the frequency of current spikes by 50%, consistent with accelerated exocytosis. This effect of cAMP was blocked if the Epac inhibitor ESI-09 was present (*Figure 3b–c*). The current spikes (see *Figure 3a*, right) reflect nucleotide release kinetics during individual exocytosis events. In the presence of cAMP, but not cAMP + ESI-09, they were markedly widened as indicated by on average 20% longer half-widths (*Figure 3d*), 30% longer decay constants (τ, *Figure 3e*), and 40% slower rising phases (25–75% slope, *Figure 3f*), compared with control. This indicates that nucleotide release is slowed by cAMP, likely because of changed fusion pore kinetics. Since the effect is blocked by ESI-09, we conclude that the cAMP effect probably is mediated by Epac.

## cAMP-dependent fusion pore regulation is absent in Epac2$^{-/-}$ (*Rapgef4$^{-/-}$*) β-cells

Since ESI-09 blocks all Epac isoforms (*Zhu et al., 2015*), we characterized fusion pore behavior in isolated β-cells from Epac2$^{-/-}$ (*Rapgef$^{-/-}$*) mice that lack all splice variants of Epac2 (*Kopperud et al., 2017*). Cells from WT or Epac2$^{-/-}$ mice were infected with adenovirus encoding the granule marker NPY-tdmOrange2 and challenged with 75 mM K$^+$ (*Figure 4a–b*). In the absence of forskolin, exocytosis was significantly slower in Epac2$^{-/-}$ cells than WT cells, and the fraction of flash-associated exocytosis events was five-fold lower (*Figure 4c–e*). This was paralleled by strikingly shorter fusion pore life-times in Epac2$^{-/-}$ cells compared with WT (*Figure 4f*). The data suggest that Epac2 is partially activated in these conditions, consistent with elevated cAMP levels in mouse β-cells in hyperglycemic conditions (*Dyachok et al., 2008*). As expected, forskolin increased both exocytosis (*Figure 4e*) and the fraction of flash events (*Figure 4c*) of WT cells. In contrast, forskolin failed to accelerate exocytosis in Epac2$^{-/-}$ cells, and the fraction of flash events was similar with or without forskolin (*Figure 4c,f–g*). We conclude therefore that the effects of cAMP on fusion pore behavior are mediated specifically by Epac2.

## Sulfonylureas delay fusion pore expansion through the same pathway as cAMP

Sulfonylureas have been reported to activate Epac (*Zhang et al., 2009*), in addition to their classical role that involves the sulfonylurea receptor (SUR). We therefore tested the effect of sulfonylureas on fusion pore behavior. INS-1 cells expressing NPY-tdmOrange2 were tested with three types of sulfonylureas, with different relative membrane permeability (tolbutamide < glibenclamide < gliclazide). In addition, diazoxide (200 µM) was present to prevent electrical activity. Exocytosis was not observed under these conditions, but could be triggered by local application of elevated K$^+$ (75 mM). In the absence of fsk, the sulfonylureas accelerated K$^+$-stimulated exocytosis about twofold over that observed in control (*Figure 5b*, left), which is consistent with earlier findings that sulfonylureas augment insulin secretion via intracellular targets (*Barg et al., 1999*). This effect was entirely due to an increase in flash-associated exocytosis events (*Figure 5b–c*) and the average NPY release time increased accordingly in the presence of sulfonylurea (*Figure 5d*). Fsk strongly stimulated both flash-associated and full fusion exocytosis in absence of sulfonylurea (*Figure 5b–c*, middle); under these conditions, sulfonylureas tended to decrease full-fusion exocytosis without effect on the frequency of flash-associated events (*Figure 5b*, middle). Accordingly, NPY release times were elevated compared with control (no fsk), and only marginally longer than with fsk alone (*Figure 5d*, right). Similar results were obtained in human β-cells, where glibenclamide increased exocytosis in the absence of fsk (p=0.01, n = 13 cells from four donors) but not in its presence (p=0.80, n = 7 cells from four donors; data not shown). The data indicate that sulfonylureas restrict fusion pore expansion through the same intracellular pathway as cAMP, which may counteract their stimulating effect on exocytosis by preventing or delaying peptide release.

   Sulfonylureas also bind to SUR1 in the plasma membrane, which leads to rapid closure of K$_{ATP}$ channels, depolarization and exocytosis. We tested the involvement of SUR1 by applying sulfonylureas acutely, which is expected to activate SUR1 in the plasma membrane but not Epac the cytosol (*Figure 5e*). Reduced diazoxide (50 µM) prevented glucose-dependent exocytosis but still allowed acute stimulation of exocytosis by sulfonylureas. Under these conditions, the fraction of flash-associated exocytosis events (*Figure 5f–g*) and the NPY release times (*Figure 5h*) were similar to control (stimulation with elevated K$^+$) for all three sulfonylureas. Taken together, the data suggest that sulfonylureas must enter the cytosol to affect fusion pore behavior, and that this effect is not mediated by the plasma membrane SUR. We excluded the possibility that sulfonylureas affect the fluorescence signal indirectly, by altering granule pH (*Figure 5—figure supplement 1*). Moreover, an EGFP-tagged SUR1 (EGFP-SUR1) expressed in INS-1 cells did not localize to exocytosis sites or affect fusion pore behavior (*Figure 5—figure supplement 2*). We therefore conclude that sulfonylureas affect fusion pore behavior through Epac2.

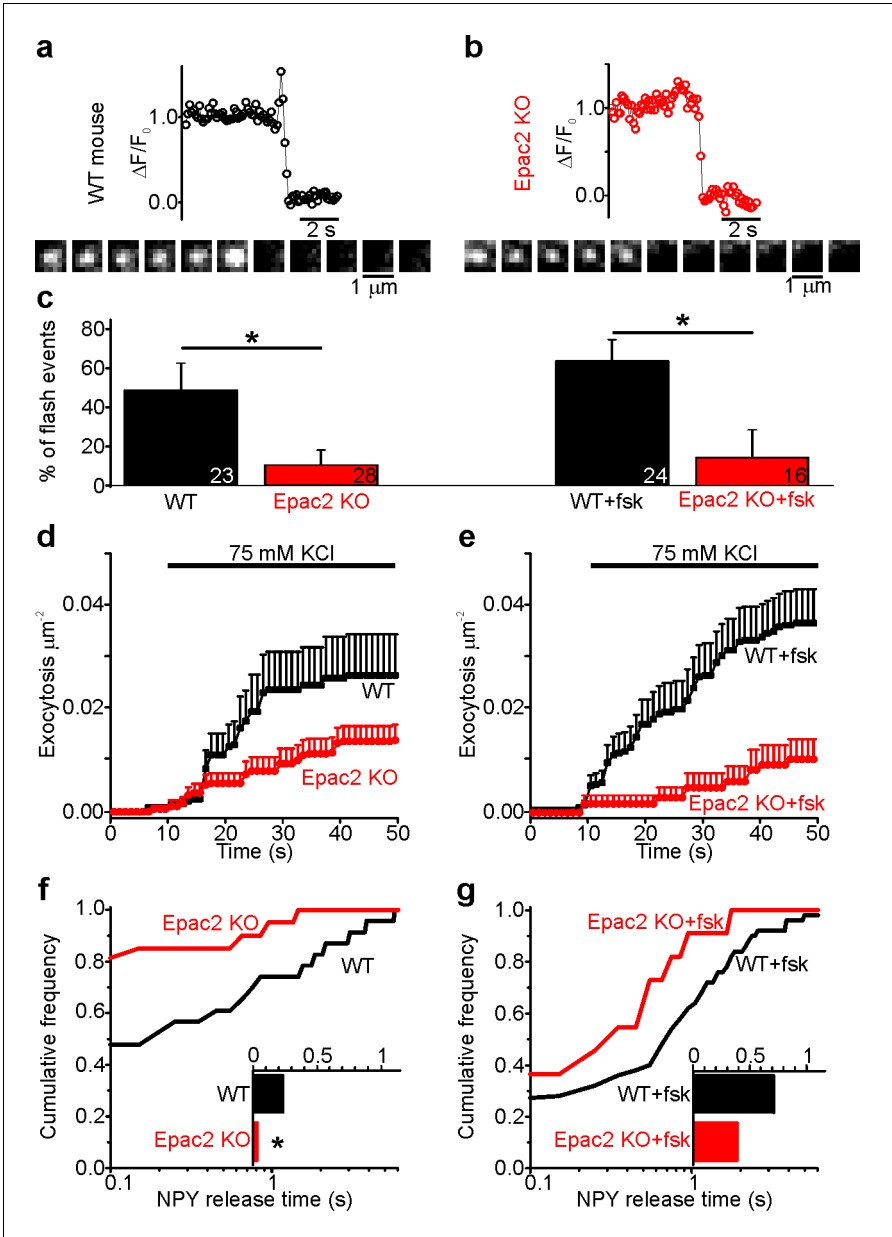

**Figure 4.** Fusion pores expand rapidly in Epac2$^{-/-}$ (*Rapgef4*$^{-/-}$) mice. (**a–b**) Examples of NPY-tdmOrange2 exocytosis events in β-cells from Epac2$^{-/-}$ mice or from wildtype littermates, stimulated with 75 mM K$^+$ in presence of forskolin. Note absence of a flash in Epac2 ko. (**c**) Fraction of flash events for experiments in (**a–b**); differences are significant in absence (p=0.027, Kruskal Wallis/Dunn test) or presence of fsk (p=0.011). Number of mice: 4 (WT); 4 (Epac2 KO); 5 (WT + fsk); 2 (Epac2 KO + fsk). n, number of cells. (**d**) Cumulative exocytosis for experiments in absence of forskolin (a,c left) for wildtype (black) and Epac2$^{-/-}$ cells (red), differences are n.s. (**e**) Cumulative exocytosis for experiments in presence of forskolin (b,c right) for wildtype (black) and Epac2$^{-/-}$ cells (red). p=0.003, Kruskal Wallis/Dunn test. (**f–g**) Cumulative frequency histograms and medians (inset) of NPY release times for exocytotic events in d (no forskolin, 23 events for wt, 22 for Epac2$^{-/-}$) and E (with forskolin, 50 events for wt, nine for Epac2-/-). Differences in f are significant (p=0.043; Kolmogorov-Smirnov test).
DOI: https://doi.org/10.7554/eLife.41711.007

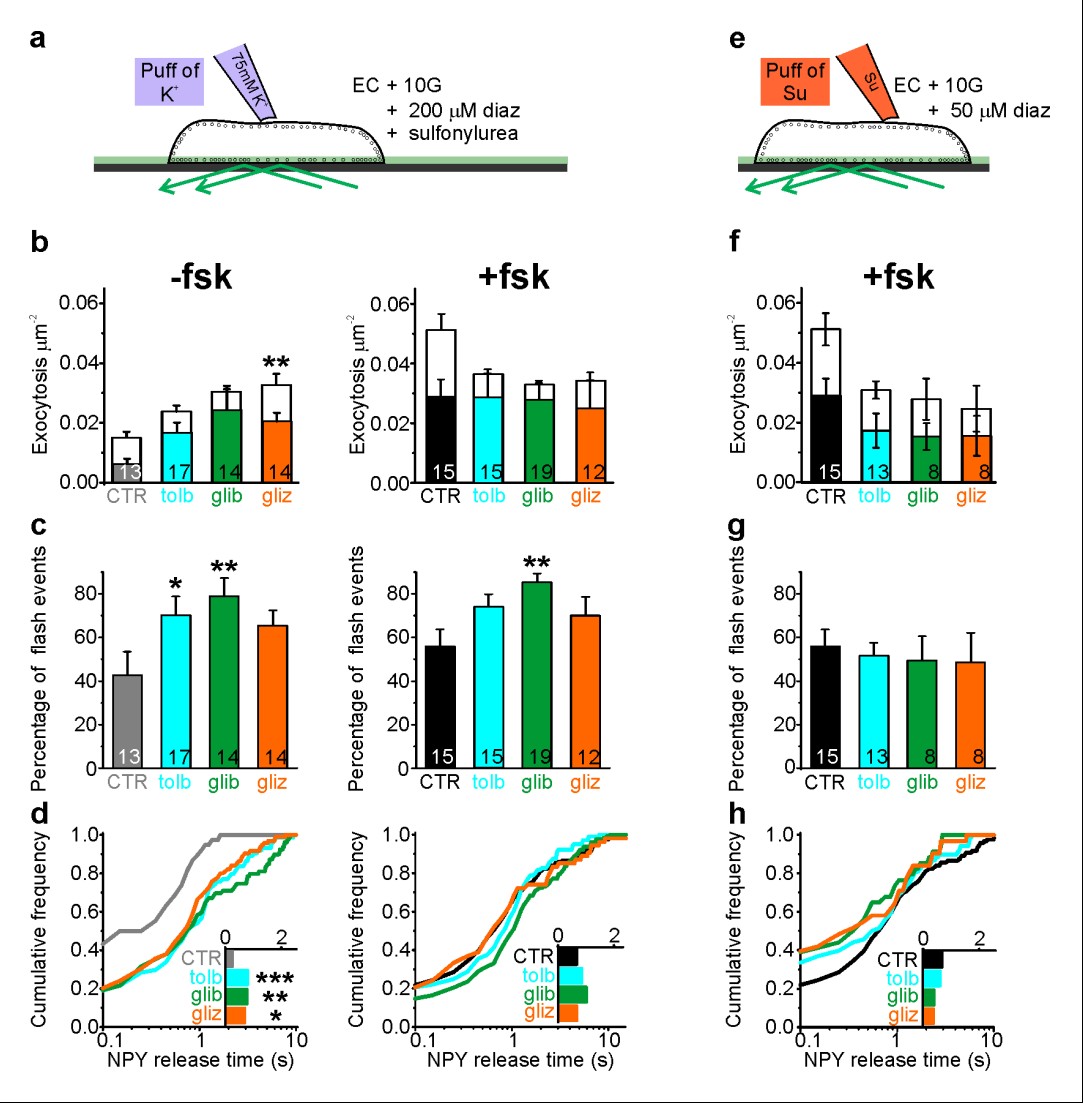

**Figure 5.** Sulfonylureas cause fusion pore restriction. (**a**) Cartoon of the experimental design in (**b–d**). INS-1 cells expressing NPY-tdmOrange2 were bathed in 10 mM glucose, diazoxide (200 µM) and either 200 µM tolbutamide (tolb), 50 µM glibenclamide (glib) or 50 µM gliclizide (gliz); exocytosis was evoked by acute exposure to 75 mM K⁺. (**b**) Exocytosis in absence (left) or presence (right) of fsk (2 µM) for flash events (color) and full fusions (white). Total exocytosis was increased by sulfonylurea in absence of fsk (p=0.15 tolb; p=0.05 glib, p=0.005 gliz, Kruskal Wallis/Dunn test vs ctrl/no fsk), but not in its presence (p=0.23 tolb; p=0.16 glib, p=0.10 glic). Sulfonylurea reduced full fusion events in presence of fsk (p=0.0045 tolb, p=0.00032 glib, 0.022 gliz, t-test). n of preps: 4 (CTR); 3 (tolb); 2 (glib); 3 (gliz); 5 (CTR + fsk); 3 (tolb + fsk); 3 (glib + fsk); 2 (gliz + fsk). n, number of cells. (**c**) Fraction of flash events for experiments in (**b**); Kruskal-Wallis/Dunn Test against ctrl/no fsk: p=0.015 tolb, p=0.001 glib, p=0.097 gliz, and against control +fsk: p=0.07 tolb; p=0.002 glib; p=0.14 gliz;); n, number of cells. (**d**) Cumulative frequency histograms and medians (insets) of NPY release times for (**b–c**). Differences vs control are significant in the absence of fsk: p=9.1*10⁻⁴ tolb, p=0.003 glib, p=0.015 gliz, Kolmogorov-Smirnov test). Insets show NPY release times for 38 (CTR), 74 (tolb), 79 (glib), 95 (gliz) events and inset on the right for 111 (CTR), 104 (tolb), 127 (glib) and 54 (gliz) events in presence of fsk. (**e**) Cartoon of the experimental design in (**f–h**). Cells were bathed in 10 mM glucose, 2 µM fsk, 50 µM diazoxide and acutely exposed to sulfonylureas (500 µM tolb, 100 µM glib or 100 µM gliz) during the recording period. (**f**) Exocytosis in presence of fsk (2 µM) for flash events (color) and full fusions (white). Differences are not significant (p=0.16 Kruskal Wallis test). n, number of cells. (**g**) Fraction of flash events for experiments in (**f**). Differences are not significant (p=0.98 Kruskal Wallis test). (**h**) Cumulative frequency histograms and medians (inset) of NPY release times for (**f–g**). Inset shows NPY release times for 111 (CTR), 68 (tolb), 34 (glib) and 31 (gliz) events.

DOI: https://doi.org/10.7554/eLife.41711.008

*Figure 5 continued on next page*

*Figure 5 continued*

The following figure supplements are available for figure 5:

**Figure supplement 1.** Granule pH is unchanged by forskolin or tolbutamide and does not affect pore lifetime.
DOI: https://doi.org/10.7554/eLife.41711.009

**Figure supplement 2.** Activation of SUR1 by tolbutamide does not affect fusion pore restriction.
DOI: https://doi.org/10.7554/eLife.41711.010

## Dynamin and amisyn-controlled restriction of the fusion pore is cAMP-dependent

The proteins dynamin and amisyn have previously been implicated in fusion pore regulation in β-cells (*Tsuboi et al., 2004*; *Collins et al., 2016*). To understand how these proteins behave around the release site, we expressed EGFP-tagged dynamin1 (*Figure 6a*) or mCherry-tagged amisyn (*Figure 6b*) together with a granule marker in INS-1 cells, and stimulated exocytosis with elevated K$^+$. In the presence of fsk, both of the two fluorescent proteins were recruited to the granule site during membrane fusion (*Figure 6c,f*, and *Figure 6—figure supplement 1*). Expression of both proteins was about 2–4 fold compared with endogenous levels (*Figure 6—figure supplement 2*), and markedly increased the NPY release times (*Figure 6d, g*) and flash-associated exocytosis events (*Figure 6e,h*). Addition of the Epac inhibitor ESI09 prevented recruitment of both dynamin1 and amisyn during flash events and reduced flash events and NPY release times below control (*Figure 6c–h*). In the absence of fsk, expression of the two proteins had no effect on fusion pore behavior, and only amisyn (but not dynamin1) was recruited to the exocytosis site (*Figure 6i–n*). When Epac was activated with S223 (no fsk), dynamin1 and amisyn were recruited during flash events, and NPY release times and flash events were increased for both proteins (*Figure 6i–n*). The data suggest that dynamin1 and amisyn are acutely recruited to the exocytosis site, where they participate in cAMP-dependent fusion pore restriction.

## Discussion

cAMP-dependent signaling restricts fusion pore expansion and promotes kiss-and-run exocytosis in β-cells (*Hanna et al., 2009*) and neuroendocrine cells (*Calejo et al., 2013*; *Machado et al., 2001*) (but see *Hatakeyama et al., 2006*). We show here that the cAMP-mediator Epac2 orchestrates these effects by engaging dynamin and perhaps other endocytosis-related proteins at the release site (*Figure 7*). Since the fusion pore acts as a molecular sieve, the consequence is that insulin and other peptides remain trapped within the granule, while smaller transmitter molecules with para- or autocrine function are released (*Obermüller et al., 2005*; *MacDonald et al., 2006*; *Taraska et al., 2003*; *Leclerc et al., 2004*). Incretin signaling and Epac activation therefore delays, or altogether prevents insulin secretion from individual granules, while promoting paracrine intra-islet communication that is based mostly on release of small transmitter molecules.

Paradoxically, two clinically important classes of antidiabetic drugs, GLP-1 analogs and sulfonylureas, activate Epac in β-cells and caused restriction of the fusion pore. Sulfonylureas have long been known to stimulate insulin secretion by binding to SUR1, which results in closure of K$_{ATP}$ channels and depolarization (*Henquin, 2000*). The drugs also accelerate PKA-independent granule priming in β-cells, which may involve activation of intracellularly localized SUR1 (*Eliasson et al., 2003*). Our data indicate that sulfonylureas exert a third mode of action that leads to the restriction of the fusion pore and therefore limits insulin release. Two pieces of evidence suggest that SUR1 is not involved in the latter. First, acute exposure to sulfonylureas had no effect on fusion pore behavior, although it blocks K$_{ATP}$ channels (indicating SUR1 activation). Only long-term exposure to sulfonylurea resulted in restricted fusion pores, likely because it allowed the drugs to enter the cytoplasm. Second, we could not detect enrichment of SUR1 at the granule release site, which precludes any direct role of the protein in fusion pore regulation. Sulfonylurea compounds have been shown to allosterically stabilize the cAMP-dependent activation of Epac (*Takahashi et al., 2013*; *Herbst et al., 2011*). Our finding that sulfonylurea caused fusion pore restriction in the absence of forskolin indicates that basal cAMP concentrations are sufficient for this effect. Since gliclazide binds the CNB1 domain without activating it (*Takahashi et al., 2013*) and still restricts the fusion pore, Epac localization at the

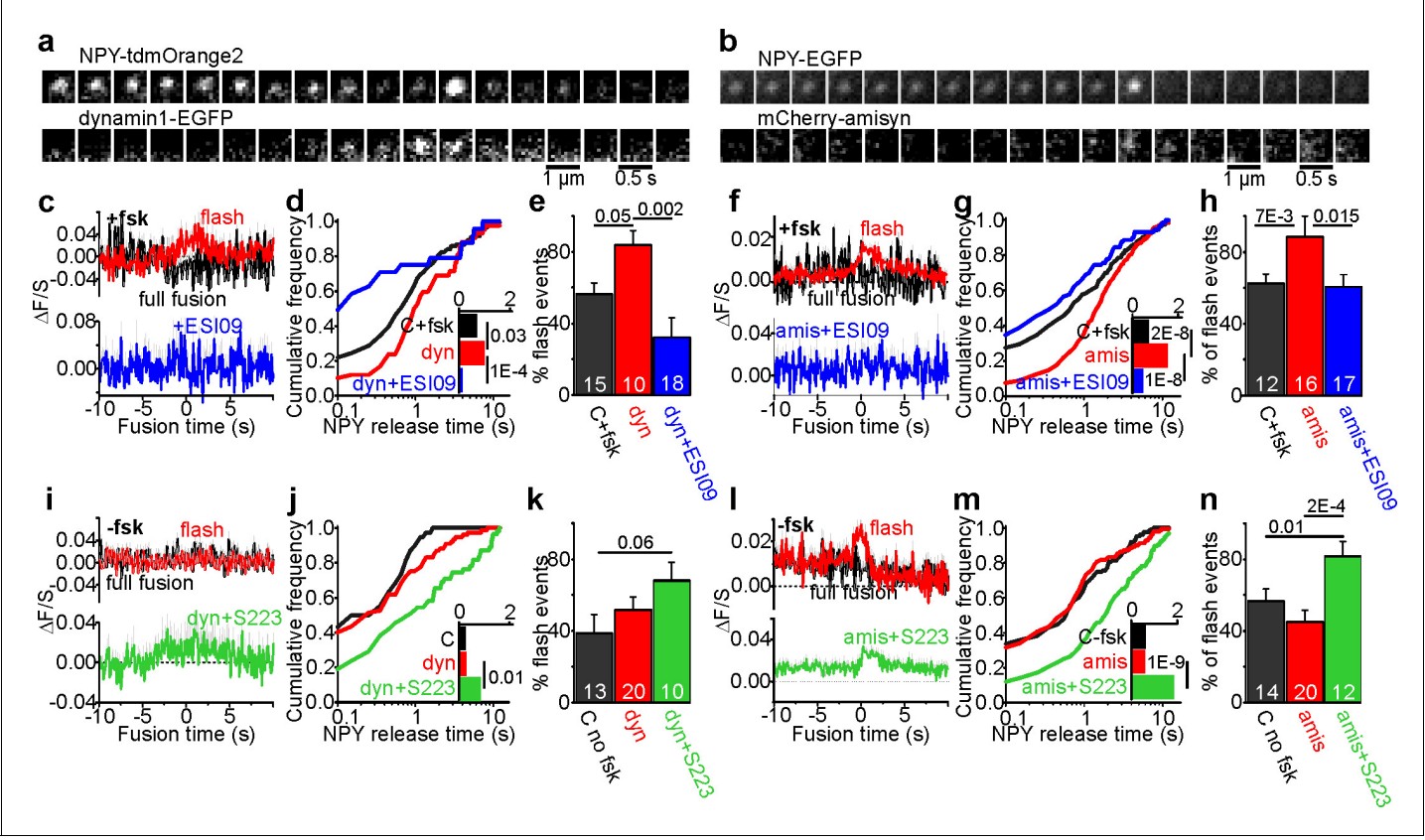

**Figure 6.** Fusion pore regulation by dynamin1 and amisyn is cAMP-dependent. (**a–b**) Example image sequence of transient recruitment of dynamin1-GFP (a, lower) or mCherry-amisyn (b, lower) to granules (upper, labeled with NPY-tdmOrange2 or NPY-EGFP) during K$^+$-stimulated exocytosis in presence of forskolin. (**c**) Average time course (± SEM) of dynamin1-GFP (dyn) fluorescence during 34 flash-type exocytosis events (red) and eight full-fusion type events (black) in presence of forskolin; and nine flash events in presence of fsk + ESI09 (blue); data points represent average of five frames and time is relative to the flash onset in the granule signal. (**d**) Cumulative frequency histograms and medians (inset, with p for Kolmogorov-Smirnov test) of NPY release times in presence of fsk in cells expressing dynamin1-EGFP (red), dynamin with added ESI09 (blue) or control (black). 119 (CTR), 42 (dyn), 24 (dyn + ESI09) events. n of preps: 5 (C + fsk); 1 (dyn); 2 (dyn + ESI-09). (**e**) Fraction of flash events in (**d**). n, number of cells, p for Kruskal-Wallis/ Dunns test. (**f**) Average time course (± SEM) of mCherry-amisyn (amis) fluorescence (red n = 274 flash events; black n = 46 full fusion events) or in presence of fsk + ESI09 (blue; n = 56 flash events). (**g**) Cumulative frequency histograms and medians (inset, with Kolmogorov-Smirnov test) of NPY release times in cells expressing mCherry-amisyn, amysin with ESI09, or control; fsk was present. 213 (CTR), 320 (amisyn), and 90 (amis +ESI09) events. n of preps: two for each. (**h**) Fraction of flash events in (**g**); p for Kruskal-Wallis/Dunn test. n, number of cells. (**i**) As in c, but without forskolin for control (black), dynamin (red), and dynamin with S223 (green); n = 37 flash events, n = 39 full fusion events for dyn and n = 40 flash events for dyn +S223. (**j–k**) As in (**d–e**), but for 38 (ctrl, black), 76 (dynamin1, red) and 55 (Dyn + S223, green) events in the absence of forskolin. n of preps: 4 (C-fsk); 2 (dyn); 2 (dyn + S223). (**l**) As in f, but without forskolin present; 65 flash events (red) and 73 full fusion events (black) for amisyn, and 154 flash events for amisyn + S223 (green). (**m–n**) As in (**g–h**), but for 123 (ctrl, black), 138 (amisyn, red) and 174 (amis + S223, green) events in the absence of forskolin. n of preps: 1 (C-fsk); 2 (amis); 2 (amis + S223).

DOI: https://doi.org/10.7554/eLife.41711.011

The following figure supplements are available for figure 6:

**Figure supplement 1.** NPY and amisyn/dynamin1 recruitment profiles at the point of release.
DOI: https://doi.org/10.7554/eLife.41711.012

**Figure supplement 2.** Quantification of overexpression.
DOI: https://doi.org/10.7554/eLife.41711.013

granule site (*Alenkvist et al., 2017*) may be enough to regulate the downstream proteins (e.g. dynamin and amisyn). It can further be speculated that the competing stimulatory (via exocytosis) and inhibitor effects (via the fusion pore) of sulfonylureas on insulin secretion, contribute to the reduction in sulfonylurea effectiveness with time of treatment. Long-term treatment with GLP-1 analogs disturbs glucose homeostasis (*Abdulreda et al., 2016*), and combination therapy of sulfonylurea and

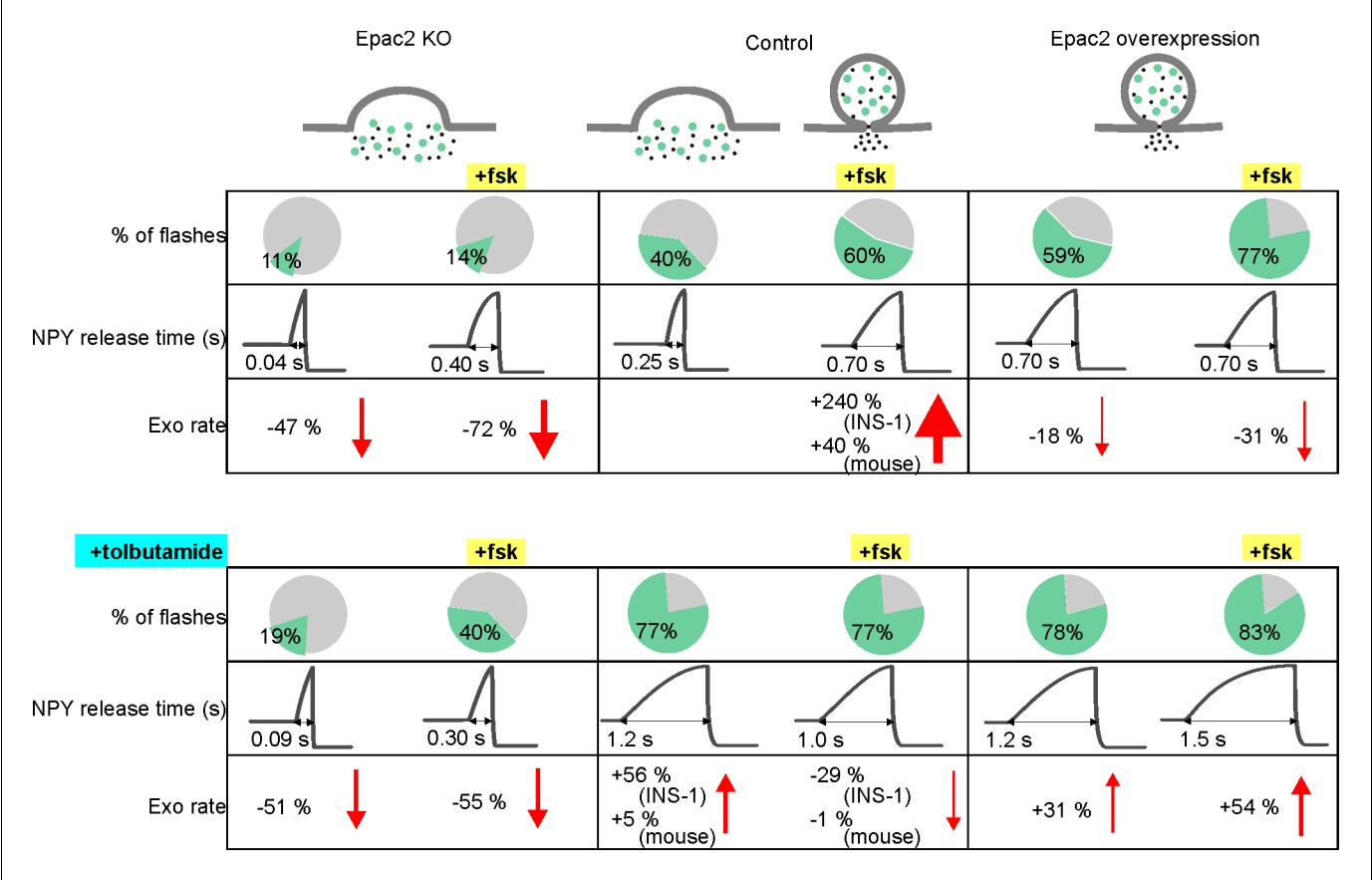

**Figure 7.** Summary of fusion pore characteristics. Fraction of events with restricted fusion pores, NPY release time and exocytosis rate for Epac2 KO (first column), controls (second column) and with Epac2 overexpression (third column) in absence (upper rows) and presence of tolbutamide (bottom rows). Changes in exocytosis are compared to controls without (left half columns) or with (right half columns) forskolin. See *Figure 7—source data 1* for details.

DOI: https://doi.org/10.7554/eLife.41711.014
The following source data is available for figure 7:

**Source data 1.**

DOI: https://doi.org/10.7554/eLife.41711.015

DPP4 inhibitors (that elevate cAMP) has been shown to lead to severe hypoglycemia (*Yabe and Seino, 2014*), an effect that likely depends on Epac (*Takahashi et al., 2015*).

Epac mediates the PKA-independent stimulation of exocytosis by cAMP (*Seino et al., 2009*) and our data suggests it may affect both priming and fusion pore restriction. This effect is rapid (*Eliasson et al., 2003*), suggesting that Epac is preassembled at the site of the secretory machinery. Indeed, Epac concentrates at sites of docked insulin granules (*Alenkvist et al., 2017*), and forms functionally relevant complexes with the tethering proteins Rim2 and Piccolo (*Fujimoto et al., 2002*). However, the amount of Epac2 present at individual release sites did not correlate with fusion pore behavior, which may indicate that the protein acts indirectly by activating or recruiting other proteins. Indeed, we show here that recruitment of two other proteins, dynamin and amisyn, depends on cAMP and Epac. Other known targets of Epac are the small GTPases Rap1 and R-Ras, for which Epac is a guanine nucleotide exchange factor (GEF). Rap1 is expressed on insulin granules and affects insulin secretion both directly (*Shibasaki et al., 2007*), and by promoting intracellular $Ca^{2+}$-release following phospholipase-C activation (*Dzhura et al., 2011*). R-Ras is an activator of phosphoinositide 3-kinase (*Marte et al., 1997*). By altering local phosphoinositide levels, Epac could therefore indirectly affect exocytosis via recruitment of C2-domain proteins such as Munc13

(*Kang et al., 2006*), and fusion pore behavior by recruitment of the PH-domain containing proteins dynamin and amisyn (*Ramachandran and Schmid, 2008*; *Abbineni et al., 2018*).

An unresolved question is whether pore behavior is controlled by mechanisms that promote pore dilation, or that instead prevent it. Dynamin causes vesicle fission during clathrin-dependent endocytosis (*Marks et al., 2001*), and since dynamin is present at the exocytosis site and required for the kiss-and-run mode (*Jackson et al., 2015*; *Tsuboi et al., 2004*; *Trexler et al., 2016*), it may have a similar role during transient exocytosis. An active scission mechanism is also suggested by the finding that granules loose some of their membrane proteins during transient exocytosis (*Tsuboi et al., 2004*; *Perrais et al., 2004*). Capacitance measurements have shown that fusion pores initially flicker with conductances similar to those of large ion channels, before expanding irreversibly (*Lollike et al., 1995*). This could result from pores that are initially stabilized through unknown protein interactions and that eventually give way to uncontrolled expansion. However, scission mechanisms involving dynamin can act even when the pore has dilated considerably beyond limit of reversible flicker behavior (*Shin et al., 2018*; *Taraska and Almers, 2004*; *Zhao et al., 2016*; *Anantharam et al., 2011*), and even relatively large granules retain their size during fusion-fission cycles (*MacDonald et al., 2006*; *Lollike et al., 1995*). Separate mechanisms may therefore operate, one that prevents pore dilation by actively causing scission, similar to the role of dynamins in endocytosis, and another by shifting the equilibrium between the open and closed states of the initial fusion pore. Curvature-sensitive proteins are particularly attractive for such roles since they could accumulate at the neck of the fused granule; such ring-like assemblies that have indeed been observed for the $Ca^{2+}$-sensor synaptotagmin (*Wang et al., 2001*). Active pore dilation has also been proposed to be driven by crowding of SNARE proteins (*Wu et al., 2017*) and α-synuclein (*Logan et al., 2017*).

β-cell granules contain a variety of polypeptides (insulin, IAPP, chromogranins) and small molecule transmitter molecules (GABA, nucleotides, 5HT) that have important para- and autocrine functions within the islet (*Braun et al., 2012*; *Caicedo, 2013*). Insulin modulates its own release by activating β-cell insulin receptors (*Leibiger et al., 2008*), stimulates somatostatin release (*Vergari et al., 2019*), and inhibits glucagon secretion (*Ravier and Rutter, 2005*). Insulin secretion is also inhibited by IAPP/amylin and chromogranin cleavage products such as pancreastatin (*Braun et al., 2012*). Of the small transmitters, GABA inhibits glucagon secretion from α-cells (*Rorsman et al., 1989*) and enhances insulin secretion (*Soltani et al., 2011*), and tonic GABA signaling is important for the maintenance of β-cell mass (*Soltani et al., 2011*). Adenine nucleotides cause β-cell depolarization, intracellular $Ca^{2+}$-release and enhanced insulin secretion (*Khan et al., 2014*; *Jacques-Silva et al., 2010*), but also negative effects have been reported (*Salehi et al., 2005*; *Poulsen et al., 1999*). Paracrine purinergic effects also coordinate $Ca^{2+}$ signaling among β-cells (*Hellman et al., 2004*), stimulate secretion of somatostatin from δ-cells (*Bertrand et al., 1990*), and target islet vasculature and macrophages as part of the immune system (*Weitz et al., 2018*). By selectively allowing small molecule release, Epac/cAMP-dependent fusion pore restriction is expected to alter both the timing and the relative volume of peptidergic vs. transmitter signaling. Given that granule priming and islet electrical activity are regulated on a second time scale, even small delays between these signals can be envisioned to affect the ratio of insulin to glucagon secretion. As illustrated by the recent finding of altered fusion pore behavior in type-2 diabetes (*Collins et al., 2016*), Epac-dependent fusion pore regulation may have profound consequences for islet physiology and glucose metabolism in vivo.

## Materials and methods

### Key resources table

| Reagent or Resource | Designation | Source of Reference | Identifiers | Additional Information |
|---|---|---|---|---|
| Strain, strain background (Adenovirus) | NPY-Venus | P Rorsman (Oxford) | | |
| Strain, strain background (Adenovirus) | NPY-tdmOrange2 | this paper | | See Constructs in Materials and methods |

*Continued on next page*

*Continued*

| Reagent or Resource | Designation | Source of Reference | Identifiers | Additional Information |
|---|---|---|---|---|
| Genetic reagent (*Mus musculus*) | *Rapgef4* KO and WT | (*Kopperud et al., 2017*) | | |
| Cell line (*Rattus norvegicus domesticus*) | INS-1 Clone 832/12 | (*Hohmeier et al., 2000*) | RRID:CVCL_7226 | H Mulder (Malmö) |
| Transfected construct (*Mus musculus*) | EGFP-Epac2 | (*Idevall-Hagren et al., 2013*) | 1068 | |
| Transfected construct (*Homo sapiens*) | NPY-tdmOrange2 | (*Gandasi et al., 2015*) | 1140 | |
| Transfected construct (*Rattus norvegicus*) | P2X$_2$-mRFP1 | (*Obermüller et al., 2005*) | 1226 | |
| Transfected construct (*Homo sapiens*) | NPY EGFP mCherry | this paper | | See Constructs in Materials and methods |
| Transfected construct (*Homo sapiens*) | Cherry2-amisyn | This paper | NM_001351940.1; 1286 | See Constructs in Materials and methods |
| Transfected construct (*Homo sapiens*) | dynamin1-GFP | W Almers (Portland) | 1342 | |
| Transfected construct (*Homo sapiens*) | NPY EGFP | W Almers (Portland) | 1008 | |
| Biological sample (*Homo sapiens*) | Human pancreatic islets | (*Goto et al., 2004*) | | Nordic Network for Clinical Islet Transplantation Uppsala |
| Antibody | Rabbit polyclonal anti-amisyn | ab153974 abcam | | 1/50 |
| Antibody | Rabbit monoclonal anti-dynamin1 | ab52852 abcam | PMID:28171750 | 1/50 |
| Chemical compound, drug | Cell dissociation buffer | Thermo Fisher | 13150016 | |
| Chemical compound, drug | Trypsin solution | Thermo Fisher | 12604–021 | |
| Chemical compound, drug | Lipofectamine 2000 | Thermo Fisher | 11668–019 | |
| Chemical compound, drug | Forskolin; Fsk | Sigma-Aldrich | F6886 | |
| Chemical compound, drug | Polylysine | Sigma-Aldrich | P5899 | |
| Chemical compound, drug | Exendin-4; Ex4 | Anaspec (Fremont CA) | AS-24463 | |
| Chemical compound, drug | Diazoxide | Sigma-Aldrich | D9035 | |
| Chemical compound, drug | BSA | Sigma-Aldrich | F0804 | |
| Chemical compound, drug | RPMI 1640 | SVA | 992680 | |
| Chemical compound, drug | L-Glutamine | Hyclone | SH30034.01 | |
| Chemical compound, drug | Tolbutamide; tolb | Sigma-Aldrich | 64-77-7 | |
| Chemical compound, drug | Glibenclamide; glib | Hoechst | | |
| Chemical compound, drug | Gliclizide; gliz | Sigma-Aldrich | 21187-98-4 | |

*Continued on next page*

*Continued*

| Reagent or Resource | Designation | Source of Reference | Identifiers | Additional Information |
|---|---|---|---|---|
| Chemical compound, drug | S223 | Biolog | B 056–01 | |
| Software, algorithm | MetaMorph | Molecular Devices | | |

## Cells

Human islets were obtained from the Nordic Network for Clinical Islet Transplantation Uppsala (*Goto et al., 2004*) under full ethical clearance (Uppsala Regional Ethics Board 2006/348) and with written informed consent. Isolated islets were cultured free-floating in sterile dishes in CMRL 1066 culture medium containing 5.5 mM glucose, 10% fetal calf serum, 2 mM L-glutamine, streptomycin (100 U/ml), and penicillin (100 U/ml) at 37°C in an atmosphere of 5% $CO_2$ up to 2 weeks. Prior to imaging, islets were dispersed into single cells by gentle agitation using $Ca^{2+}$-free cell dissociation buffer (Thermo Fisher Scientific) supplemented with 10% (v/v) trypsin (0.05% Thermo Fisher Scientific). INS1-cells clone 832/13 (*Hohmeier et al., 2000*) were maintained in RPMI 1640 (Invitrogen) with 10 mM glucose, 10% fetal bovine serum, streptomycin (100 U/ml), penicillin (100 U/ml), Sodium pyruvate (1 mM), and 2-mercaptoethanol (50 µM). The ins1 832/13 cells were screened by PCR and found negative for mycoplasma.

Mouse islets were isolated from 5 to 12 months old WT and Epac2$^{-/-}$ (*Kopperud et al., 2017*) (*Rapgef4$^{-/-}$*) animals. The Epac2 deletion involves exons 12–13, which include the high-affinity cAMP binding domain present in all Epac2 isoforms, in contrast to previously reported knockout strain (*Shibasaki et al., 2007*), which only lacks the Epac2A isoform. The mice were anesthetized and the pancreas dissected out and cleared from fat and connective tissue in ice-cold Ca5 solution (in mM 125 NaCl, 5KCl, 1.2 MgCl$_2$, 1.28 CaCl$_2$, 10 HEPES; pH 7.4 with NaOH). Pancreas was injected with Collagenase P (1 mg/ml) and cut into small pieces before mechanical dissociation (7 min at 37°C). BSA was added immediately and islets were washed 3X with ice cold Ca5 buffer with BSA. Islets were dispersed into single cells using $Ca^{2+}$-free cell dissociation buffer (supplemented with 10% (v/v) trypsin) and gentle agitation. Dispersed cells were sedimented by centrifugation, resuspended in RPMI 1640 medium (containing 5.5 mM glucose, 10% fetal calf serum, 100 U/ml penicillin and 100 U/ml streptomycin).

The cells were plated onto 22 mm polylysine-coated coverslips and were transduced the next day using adenovirus (human and mouse cells) or transfected the same day with plasmids (INS1 cells, using Lipofectamine2000, Invitrogen) encoding the granule markers NPY-Venus, NPY-EGFP or NPY-tdOrange. Imaging proceeded 24–36 hr later.

## Constructs

The open-reading frame of human amisyn (NM_001351940.1) was obtained as a synthetic DNA fragment (Eurofins, Germany) and was cloned into pCherry2 C1 (Addgene, plasmid nr 54563) by seamless PCR cloning. The linker between Cherry2 and amisyn translates into the peptide SGLRSRAQASNSAV. The plasmid N1 NPY-EGFP-mCherry coding for NPY-linker(TVPRARDPPVAT)-EGFP-linker(KRSGGSGGSGGS)-mCherry was made by seamless PCR cloning. The correct open-reading frame of both Cherry2-linker-amisyn and NPY-EGFP-mCherry was confirmed by Sanger sequencing (Eurofins, Germany). The NPY-tdOrange2 adeno virus was made using the RAPAd vector system (Cell Biolabs, San Diego). NPY-tdOrange2 (*Gandasi et al., 2015*) was cloned into the pacAd5 CMVK-NpA Shuttle plasmid (Cell Biolabs). Virus was produced in HEK293 cells and isolated according to the instructions of the manufacturer (Cell Biolabs).

## Solutions

Cells were imaged in (mM) 138 NaCl, 5.6 KCl, 1.2 MgCl$_2$, 2.6 CaCl$_2$, 10 D-glucose 5 HEPES (pH 7.4 with NaOH) at 32–34°C. Exocytosis was evoked with high 75 mM K$^+$ (equimolarly replacing Na$^+$), applied by computer-timed local pressure ejection through a pulled glass capillary. For K$^+$-induced exocytosis, spontaneous depolarizations were prevented with 200 µM diazoxide (50 µM for *Figure 5e–h*). In *Figure 5e–h*, exocytosis was evoked by sulfonylureas (500 µM tolbutamide, 200 µM glibenclamide or 200 µM gliclazide). For electrophysiology, glucose was reduced to 3 mM, and the

electrodes were filled with (mM) 125 CsCl, 10 NaCl, 1.2 MgCl$_2$, 5 EGTA, 4 CaCl$_2$, 3 Mg-ATP, 0.1 cAMP, 10 HEPES (pH 7.15 using CsOH).

## Immunocytochemistry

To quantify the overexpression, INS-1 cell were transfected with either Cherry2-amisyn or Dynamin1-GFP, fixed 24 hr later in 3.8% formaldehyde in phosphate-buffered saline (PBS) for 30 min at 25°C and washed in PBS. The cells were permeabilized in 0.2% Triton X-100 in PBS for 5 min and washed in PBS. Blocking was done using 5% FBS in PBS for 1–2 hr at 25°C. Cells were then incubated with a primary antibody (anti-Dynamin1, ab52852 abcam or anti-Amisyn, ab153974 abcam) both diluted 1/50 in 5% FCS in PBS over night at 4°C and washed again in PBS. Incubation with secondary antibody (Alexa Fluor 488 anti-rabbit or Alexa Fluor 555 anti-rabbit, Invitrogen) diluted 1/1000 in 5% FCS in PBS was performed for 1 hr at 25°C and subsequently the cells were washed in PBS.

## TIRF microscopy

Human cells were imaged using a lens-type total internal reflection (TIRF) microscope, based on an AxioObserver Z1 with a 100x/1.45 objective (Carl Zeiss). TIRF illumination with a calculated decay constant of ~100 nm was created using two DPSS lasers at 491 and 561 nm (Cobolt, Stockholm, Sweden) that passed through a cleanup filter (zet405/488/561/640x, Chroma) and was controlled with an acousto-optical tunable filter (AA-Opto, France). Excitation and emission light were separated using a beamsplitter (ZT405/488/561/640rpc, Chroma) and the emission light chromatically separated (QuadView, Roper) onto separate areas of an EMCCD camera (QuantEM 512SC, Roper) with a cutoff at 565 nm (565dcxr, Chroma) and emission filters (ET525/50 m and 600/50 m, Chroma). Scaling was 160 nm per pixel.

INS1 and mouse cells were imaged using a custom-built lens-type TIRF microscope based on an AxioObserver D1 microscope and a 100x/1.45 NA objective (Carl Zeiss). Excitation was from two DPSS lasers at 473 nm and 561 nm (Cobolt), controlled with an acousto-optical tunable filter (AOTF, AA-Opto) and using dichroic Di01-R488/561 (Semrock). The emission light was separated onto the two halves of a 16-bit EMCCD camera (Roper Cascade 512B, gain setting at 3800 a.u. throughout) using an image splitter (DualView, Photometrics) with ET525/50 m and 600/50 m emission filters (Chroma). Scaling was 100 nm per pixel for INS-1 experiments and 160 nm for mouse cells. The frame rate was 10 frames*s$^{-1}$, with 100 ms exposures.

## Image analysis

Exocytosis events were identified manually based on the characteristic rapid loss of the granule marker fluorescence (most fluorescence lost within 1–2 frames) in cells which exhibited minimum of 1 event/cell (except mouse cells, where all cells were included). Events were classified as flash events if they exhibited an increase in the fluorescence signal before the rapid loss of the granule fluorescence. The NPY release times were obtained for both types of events by non-linear fitting with a discontinuous function in Origin as described previously (*Gandasi et al., 2015*). Protein binding to the release site (ΔF/S) was measured as described previously (*Gandasi and Barg, 2014*).

## Electrophysiology

ATP release was measured in INS1 cells expressing RFP-tagged P2X$_2$ receptor (*Obermüller et al., 2005*). Cells were voltage-clamped in whole-cell mode using an EPC-9 amplifier and PatchMaster software (Heka Elektronik, Lambrecht, Germany) with patch-clamp electrodes pulled from borosilicate glass capillaries that were coated with Sylgard close to the tips, and fire-polished (resistance 2–4 MΩ). The free [Ca$^{2+}$] was calculated to be 600 nM (WEBMAXC standard) and elicited exocytosis that was detected as P2X$_2$-dependent inward current spikes. Currents were filtered at 1 kHz and sampled at 5 kHz. Spike analysis was performed using automated program for amperometric recordings in IGOR Pro (*Segura et al., 2000*), with the threshold set at eight times the RMS noise during event-free section of recording.

## Statistics

Data are presented as mean ± SEM unless otherwise stated. Statistical significance was tested (unless otherwise stated) and is indicated by asterisks (*p<0.05, **p<0.01, ***p<0.001). The not

normally distributed exocytosis rates and ratios of flash events were tested with Kruskal Wallis with post hoc Dunn test and NPY release times were tested with Kolmogorov-Smirnov test.

## Acknowledgements

We thank J Saras, P-E Lund, Y Xu and A Thonig (Uppsala University) for expert technical assistance, and D Machado (University of La Laguna) for spike analysis software.

## Additional information

### Funding

| Funder | Grant reference number | Author |
| --- | --- | --- |
| Family Ernfors Foundation | | Alenka Guček<br>Anders Tengholm<br>Sebastian Barg |
| Uppsala Universitet | Olga Jönssons stipend | Alenka Guček |
| P O Zetterlingsstiftelse | | Alenka Guček |
| European Foundation for the Study of Diabetes | | Nikhil R Gandasi<br>Anders Tengholm<br>Sebastian Barg |
| Swedish Society for Medical Research | | Nikhil R Gandasi |
| Novo Nordisk | | Nikhil R Gandasi<br>Anders Tengholm<br>Sebastian Barg |
| Norwegian Research Council | | Marit Bakke |
| Helse-Bergen | | Marit Bakke |
| Swedish Research Council | 2014-02575 | Anders Tengholm<br>Sebastian Barg |
| Diabetes Wellness Network Sweden | | Anders Tengholm<br>Sebastian Barg |
| Swedish Diabetes Society | | Anders Tengholm<br>Sebastian Barg |
| Exodiab network | | Anders Tengholm<br>Sebastian Barg |
| Swedish Research Council | 2017-00956 | Anders Tengholm<br>Sebastian Barg |
| Swedish Research Council | 2018-02871 | Anders Tengholm<br>Sebastian Barg |
| Hjärnfonden | | Sebastian Barg |

The funders had no role in study design, data collection and interpretation, or the decision to submit the work for publication.

### Author contributions

Alenka Guček, Conceptualization, Data curation, Formal analysis, Validation, Investigation, Visualization, Writing—original draft, Writing—review and editing; Nikhil R Gandasi, Conceptualization, Formal analysis, Investigation, Writing—review and editing; Muhmmad Omar-Hmeadi, Methodology, Writing—review and editing; Marit Bakke, Resources, Funding acquisition; Stein O Døskeland, Anders Tengholm, Resources, Funding acquisition, Writing—review and editing; Sebastian Barg, Conceptualization, Resources, Data curation, Supervision, Funding acquisition, Visualization, Methodology, Project administration, Writing—review and editing

Author ORCIDs
Alenka Guček (iD) https://orcid.org/0000-0003-4453-1498
Muhmmad Omar-Hmeadi (iD) http://orcid.org/0000-0001-8893-7348
Anders Tengholm (iD) http://orcid.org/0000-0003-4508-0836
Sebastian Barg (iD) https://orcid.org/0000-0003-4661-5724

## Ethics

Animal experimentation: This study was performed in strict accordance with European and Swedish legislation, fundamental ethical principles and approved by the Regional Ethics Board Uppsala (license number 31, 1-32).

## Decision letter and Author response

Decision letter https://doi.org/10.7554/eLife.41711.020
Author response https://doi.org/10.7554/eLife.41711.021

## Additional files

### Supplementary files

• Transparent reporting form
DOI: https://doi.org/10.7554/eLife.41711.016

### Data availability

Source data file has been provided for Figure 7. All raw data are available on the Dryad Digital Repository (http://dx.doi.org/10.5061/dryad.6b604g8).

The following dataset was generated:

| Author(s) | Year | Dataset title | Dataset URL | Database and Identifier |
|---|---|---|---|---|
| Gucek A, Gandasi NR, Omar-Hmeadi M, Bakke M, Døskeland SO, Tengholm A, Barg S | 2019 | Data from: Fusion pore regulation by cAMP/Epac2 controls cargo release during insulin exocytosis | http://dx.doi.org/10.5061/dryad.6b604g8 | Dryad Digital Repository, 10.5061/dryad.6b604g8 |

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
