## [Decision Letter]

Thank you for submitting your article "Fusion pore regulation by Epac2/cAMP controls cargo release during insulin exocytosis" for consideration by *eLife*. Your article has been reviewed by Vivek Malhotra as the Senior Editor, a Reviewing Editor, and three reviewers. The following individuals involved in review of your submission have agreed to reveal their identity: Ling-Gang Wu (Reviewer #1).

The reviewers have discussed the reviews with one another and the Reviewing Editor has drafted this decision to help you prepare a revised submission.

Summary:

In this work the regulation of fusion pore dynamics in pancreatic β-cells is studied. The authors use human pancreatic β-cells for key first experiments, and subsequently from INS-1 cells and the Epac2 knockout mouse. Although it has been previously reported that Epac2 is directly involved in exocytosis at the exocytotic machinery, the precise mechanisms by which Epac2 regulates insulin exocytosis are poorly understood. The results suggest that cAMP and sulfonylureas restrict the fusion pore opening thereby limiting the release of insulin. Finally, the data further suggest that dynamin and amysin are part in the fusion pore restriction and that recruitment of these proteins to the release site is cAMP-dependent. The work adds new information on the role of Epac2 in fusion pore dynamics regarding the release from β-cells of not only insulin but also many other signaling smaller nucleotides.

There are several concerns as outlined below and we outline experimental data that should be included in the revised manuscript.

Essential revisions:

The heterogeneity among cells from different human donors is wide, and it can be argued whether results will hold if the authors increase the number of donors in their study. Moreover, the data from INS-1cells (rat islet origin and not human) require considering differences between human and rodent β-cells. It is essential that the key experiments in Figure 1 and the dynamin/amysin experiments in Figure 6 are repeated in the human cell line EndoC.

The authors do not really show the link between cAMP and dynamin/amysin. What mechanisms are involved in this process? The authors should address this by perhaps a speculative model in their discussion.

Abstract, the conclusion is wrong: cAMP elevation leads to pore expansion and peptide release, but not when Epac2 is inactivated pharmacologically or in Epac2^-/-^ mice. Conversely, overexpression of Epac2 impedes pore expansion. Based on the data, the conclusion might be: cAMP elevation restricts pore and leads to a slower expansion of the fusion pore and peptide release, but not when Epac2 is inactivated pharmacologically or in Epac2^-/-^ mice. Consistently, overexpression of Epac2 impedes pore expansion.

The complete release of NPY proceeds via full collapse fusion as shown in Figure 1(B) and in the text (Introduction), but according to reference (Shin et al., 2018), the vesicles can maintain the omega-shaped structures even after complete NPY release, and the pores can constrict after the complete release. Full-collapse is not observed for dense-core vesicle fusion in chromaffin cells (Shin et al., 2018). In addition, the fusion pore does not have to be larger than 10-30 nm for complete NPY release, because EGFP size is only 3.7nm as stated. The diagram in Figure 1(B) should be modified accordingly to avoid extrapolation from a pore more than 3.7 nm to full collapse fusion, and, "fusion pore lifetime" should be changed to "NPY release time" or other proper terms both in the legends and the main text. The methods used by the authors do not detect fusion pore lifetime, but only the NPY release time course (see Shin et al., 2018 for imaging of the fusion pore lifetime).

What is the reason that authors restricted exocytosis events to the rapid loss of the granule marker fluorescent (1-2 frames)? What about granule marker loss within 3-10 frames or >10 frame? Please clarify why these events are excluded. On the other hand, in Figure 6a, the release time of the NPY after flash takes more than 2 seconds with multiple frames (obviously >4 frames). This is inconsistent with the author's criteria of exocytosis events with the rapid loss within 1-2 frames. Please show the frame rate of all experiments since the different frame rates may generate different fraction of flash events vs full fusion events.

Rapid NPY spot fluorescence decrease can also be due to vesicle undocking (docked vesicle leaving the docking site), rather than fusion as in Figure 1A. The possibility of undocking should be ruled out by control experiments. A control experiment can be co-expressing pH-sensitive vesicular protein or adding dye in the bath solution to indicate fusion.

In subsection “cAMP-dependent fusion pore restriction is regulated by Epac but not PKA”, the authors state that PKA inhibition with Rp8-Br-cAMPS decreases exocytosis ~40% but not the fraction of flash events which is shown in Figure 1F. Then the percentage of flash events should be >80% in Figure 1D for fsk+RP-8, but the flash percentage is similar as fsk alone with ~60-65%. How is this possible? This needs to be clarified to support the conclusion that Epac rather than PKA is responsible for cAMP-dependent fusion pore regulation.

At the end of Introduction, the authors write that activation of Epac2 via GLP-1 and sulfonylurea will hinder full fusion and that this is "expected to reduce their full fusion as insulin secretagogoues". However, there are no data available in this study (or in published elsewhere) showing that addition of GLP-1 or sulfonylurea will reduce insulin secretion. How do the authors validate the importance of their study? Is glucose stimulated insulin secretion decreased in the cells (Figure 2) overexpressing Epac2? What about Epac2 and insulin secretion in EndoC cells?

The efficiency of the overexpression is not shown. What is the mRNA and protein levels of Epac2 (Figure 2), amysin and dynamin (Figure 6) after overexpression?

The authors have not performed the proper statistics for the data. A simple t-test is not enough when comparing multiple conditions. At least one-way ANOVA with ad hoc-test for multiple comparison.

The authors used Epac2 KO mice established in Kopperud et al. 2017, in which the characteristics of microvascular permeability of these mice was examined. However, the characteristics regarding glucose homeostasis including insulin secretion were not well studied in the paper. Furthermore, other lines of Epac2 KO mice have already been established and analyzed. The authors need to examine at least the insulin secretory profiles in the Epac2 KO mice used in this study in vivo and in vitro, and discuss them by referring to the previous studies using other KO strains.

Figure 2 shows that the percentage of flash events is increased by overexpression of Epac2. However, the actual number of flash events is not changed but full fusion events are decreased (Figure 2B), resulting in a decrease in the total number of fusion events (Figure 2A). In Figure 1, Epac activation by S223 increases the total number of exocytotic events by increasing flash events without changing the number of full fusion events. It seems that the fusion pore-regulating mechanism is different between these two conditions. Any thoughts on this issue?

In Figure 4, the total number of exocytotic events is not increased by fsk in Epac2 KO β-cells. However, it is unlikely that cAMP cannot increase the number of exocytotic events in the absence of Epac2, as PKA signaling remains intact. What is the authors' interpretation of these results?

In Figure 4F and 4G, fsk seems to prolong fusion pore lifetime significantly in both WT and Epac2 KO β-cells, although the authors state that the lifetimes are similar with or without forskolin. This should be modified.

In Figure 5, glib, tolb and gliz show similar effects on fusion pore regulation. However, the gliclazide data are inconsistent with the previous reports in which gliclazide among various sulfonylureas does not activate Epac2 (Zhang et al., 2009, Takahashi et al., 2013, Takahashi et al., 2013). The results should be confirmed using β-cells from Epac2 KO mice.

The authors show in Figure 6 that dynamin and amisyn are translocated to the site of exocytosis in a cAMP-dependent manner, and suggest that Epac protein is preassembled at the site of the secretory machinery. The authors also mention the involvement of Epac2 in the recruitment of dynamin and amisyn to the site of exocytosis, based on the observations of the translocation of these proteins by cAMP stimulation, which do not provide sufficient evidence to draw the authors' conclusion. The authors should show the behaviors of dynamin and amisyn in Epac2 KO β-cells and when Epac is activated pharmacologically.

The significance of the restriction of fusion pore expansion is not clear. Both full fusion and flash events eventually release all contents of the secretory vesicle. Thus, the prolonged fusion pore lifetime means only a few seconds delay in insulin release compared to release of small molecules (or small molecules release a few seconds in advance of the insulin release). Although the authors discuss the roles of these small molecules in autocrine and paracrine signaling in islets, the physiological and pathophysiological relevance of the time difference in the release of insulin and small molecules should be further discussed.

It has previously been shown that there are different modes of insulin exocytosis, based on the dynamics of insulin granules (predocked granules, newcomer granules, etc.) (Shibasaki et al.,2007 and Nagamatsu et al., 2006). Is there any difference in the mode of insulin exocytosis between the present study and the previous reports? The authors might want to discuss the modes of exocytotic events found in the present study in relation to previous studies.

[Editors' note: further revisions were requested prior to acceptance, as described below.]

Thank you for resubmitting your work entitled "Fusion pore regulation by cAMP/Epac2 controls cargo release during insulin exocytosis" for further consideration at *eLife*. Your revised article has been favorably evaluated by Vivek Malhotra (Senior Editor), a Reviewing Editor, and three reviewers.

The manuscript has been improved but there are some remaining issues that need to be addressed before acceptance, as outlined below:

1) Figure 1B. The drawing of full fusion with a wide concave structure may be subject to error, as recent super-resolution STED imaging shows that fusing dense-core vesicles maintain an omega-shape till the STED resolution limit (Shin et al., 2018; Chiang et al., 2014). We suggest to re-draw the concave shape in Figure 1B as an omega shape with a larger pore than the other omega profile at the left side. This does not mean that full-collapse does not occur in β-cells, but the authors' data can only say that the fusion pore is larger than NPY-venus.

2) The reviewer's comment: “Figure 2 shows that the percentage of flash events…”

The authors do not respond to the comment properly. The reviewers asked the authors to discuss the mechanisms of the different findings between Figure 2 and Figure 1. This should be reconsidered.

3) The reviewer's comment: “In Figure 5, Glib, tolb and gliz show similar effects…”

The authors have performed the requested experiments, and suggested that Epac2 is required for the effect of gliz on the fusion pore. The result on gliz is different from the previous reports (Zhang et al., 2009; Takahashi et al., 2013). The authors should discuss reasons for the discrepancy.

4) The reviewer's comment: “The role of Eoac2(cAMP-GEFII) in insulin exocytosis…” The authors responded to the comment, but the modified statement has not been incorporated in the text (Introduction).

5) The reviewer's comment: “The source of reagents…” The source is provided, but the source of the Epac2 agonist S223 is missing.

6) The reviewer's comment: The description "inhibitors of the GLP-1 peptidase"… In the authors' response, "inhibitors of DPP-4 peptidase" should be "inhibitors of DPP-4"(delete"peptidase").

---

## [Author Response]

Summary:In this work the regulation of fusion pore dynamics in pancreatic β-cells is studied. The authors use human pancreatic β-cells for key first experiments, and subsequently from INS-1 cells and the Epac2 knockout mouse. Although it has been previously reported that Epac2 is directly involved in exocytosis at the exocytotic machinery, the precise mechanisms by which Epac2 regulates insulin exocytosis are poorly understood. The results suggest that cAMP and sulfonylureas restrict the fusion pore opening thereby limiting the release of insulin. Finally, the data further suggest that dynamin and amysin are part in the fusion pore restriction and that recruitment of these proteins to the release site is cAMP-dependent. The work adds new information on the role of Epac2 in fusion pore dynamics regarding the release from β-cells of not only insulin but also many other signaling smaller nucleotides.There are several concerns as outlined below and we outline experimental data that should be included in the revised manuscript.Essential revisions:The heterogeneity among cells from different human donors is wide, and it can be argued whether results will hold if the authors increase the number of donors in their study. Moreover, the data from INS-1 cells (rat islet origin and not human) require considering differences between human and rodent β-cells. It is essential that the key experiments in Figure 1 and the dynamin/amysin experiments in Figure 6 are repeated in the human cell line EndoC.

The EndoC line is not available to us, despite a request to the authors. During revision, we instead added data from four additional human donors, so that each of the conditions in Figure 1 now has been measured with 5-7 donors. We do not see strong donor to donor variation with respect to fusion pore behavior, and even average exocytosis is relatively similar between preps from non-diabetic donors (see Author response image 1 and Gandasi et al., 2018). While there are differences in exocytosis rate between Ins1 and human cells, the effect of forskolin is very similar between the two. Exocytosis roughly doubles, the fraction of flash events shifts from 40 to 60%, and pore lifetimes increase from 0.27 to 0.7s for Ins1 and from 0.3 to 0.9s for human cells. We would also like to point out that Ins1 is a far more established model for insulin exocytosis (at least 1800 papers in PubMed) than EndoC (about 60 papers, of which only 4 studied exocytosis).

**Author response image 1. respfig1:** K^+^-stimulated exocytosis in the indicated conditions. In (**a**) each symbol represents the average of a donor, in (**b**), each symbol represents one cell.

The authors do not really show the link between cAMP and dynamin/amysin. What mechanisms are involved in this process? The authors should address this by perhaps a speculative model in their discussion.

We now include data demonstrating that pharmacological inactivation of Epac prevents cAMP-dependent recruitment of dynamin and amisyn, and that the specific Epac activator S223 leads to recruitment of these proteins during flash exocytosis even in absence of cAMP raising agents (Figure 6).

We speculate (in the Discussion section) that cAMP/Epac controls the local generation of PIP2, which in turn leads to recruitment of the PH-domain containing proteins dynamin and amisyn:

“Indeed, we show here that recruitment of two other proteins, dynamin and amisyn, depends on cAMP and Epac. Other known targets of Epac are the small GTPases Rap1 and R-Ras, for which Epac is a guanine nucleotide exchange factor (GEF). Rap1 is expressed on insulin granules and affects insulin secretion both directly ^65^, and by promoting intracellular Ca^2+^-release following phospholipase-C activation ^66^. R-Ras is an activator of phosphoinositide 3-kinase ^67^. By altering local phosphoinositide levels, Epac could therefore indirectly affect exocytosis via recruitment of C2-domain proteins such as Munc13 ^68^, and fusion pore behavior by recruitment of the PH-domain containing proteins dynamin and amisyn ^69,70^.”

Abstract, the conclusion is wrong: cAMP elevation leads to pore expansion and peptide release, but not when Epac2 is inactivated pharmacologically or in Epac2^-/-^ mice. Conversely, overexpression of Epac2 impedes pore expansion. Based on the data, the conclusion might be: cAMP elevation restricts pore and leads to a slower expansion of the fusion pore and peptide release, but not when Epac2 is inactivated pharmacologically or in Epac2^-/-^ mice. Consistently, overexpression of Epac2 impedes pore expansion.

We thank the reviewer for pointing out this error, which has been corrected as suggested.

The complete release of NPY proceeds via full collapse fusion as shown in Figure 1(B) and in the text (Introduction), but according to reference (Shin et al., 2018), the vesicles can maintain the omega-shaped structures even after complete NPY release, and the pores can constrict after the complete release. Full-collapse is not observed for dense-core vesicle fusion in chromaffin cells (Shin et al., 2018). In addition, the fusion pore does not have to be larger than 10-30 nm for complete NPY release, because EGFP size is only 3.7nm as stated. The diagram in Figure 1(B) should be modified accordingly to avoid extrapolation from a pore more than 3.7 nm to full collapse fusion, and, "fusion pore lifetime" should be changed to "NPY release time" or other proper terms both in the legends and the main text. The methods used by the authors do not detect fusion pore lifetime, but only the NPY release time course (see Shin et al., 2018 for imaging of the fusion pore lifetime).

We agree that long-lived omega-shaped structures cannot be excluded in pancreatic β-cells. This is now acknowledged (Introduction). We have also changed the fusion pore lifetime to NPY release time thorough the figures and text, as suggested.

What is the reason that authors restricted exocytosis events to the rapid loss of the granule marker fluorescent (1-2 frames)? What about granule marker loss within 3-10 frames or >10 frame? Please clarify why these events are excluded. On the other hand, in Figure 6a, the release time of the NPY after flash takes more than 2 seconds with multiple frames (obviously >4 frames). This is inconsistent with the author's criteria of exocytosis events with the rapid loss within 1-2 frames. Please show the frame rate of all experiments since the different frame rates may generate different fraction of flash events vs full fusion events.

Detection of events was done by eye, using the criterion that the majority of the signal is lost suddenly (1-2 frames). This is now stated in the Materials and methods section, where we added the following text:

“The frame rate was 10 frames*s^-1^, with 100 ms exposures” subsection “TIRF microscopy” and “characteristic rapid loss of the granule marker fluorescence (most fluorescence lost within 1-2 frames)” subsection “Image analysis”.

In rare cases, a small portion of the signal remained at the release site (as in Figure 6A, note that only every fifth frame is shown). This is apparent in the quantification of the NPY-mOrange signal (now shown in Figure 6—figure supplement 1). There are four possible explanations for the remaining fluorescence, (1) binding of the label to the outside of the cell after complete release (Michael, 2006), (2) kiss and run followed by undocking or re-acidification (Obermüller, 2005), (3) stray light from a nearby granule that did not undergo exocytosis, or (4) rapid undocking (see answer to the next question). We would maintain that the effect of all of this on our flash/no flash ratio measurement is minor. Occasionally, pore lifetimes for no-flash events are overestimated by the fitter, but this would not affect the conclusions of the paper.

Rapid NPY spot fluorescence decrease can also be due to vesicle undocking (docked vesicle leaving the docking site), rather than fusion as in Figure 1A. The possibility of undocking should be ruled out by control experiments. A control experiment can be co-expressing pH-sensitive vesicular protein or adding dye in the bath solution to indicate fusion.

Undocking is a rare event, 25-100 times less frequently than K^+^-stimulated exocytosis (eg Gandasi et al., 2018). However, control experiments were included in two of our previous papers. Accordingly, NPY-EGFP fluorescence disappears rapidly during exocytosis, while loss of the signal during undocking takes several seconds (Gandasi and Barg,2014; Figure S7). Multiplane confocal imaging also indicated that undocking is very slow, compared with exocytosis/content release (Barg, 2002, Figure 3).

In subsection “cAMP-dependent fusion pore restriction is regulated by Epac but not PKA”, the authors state that PKA inhibition with Rp8-Br-cAMPS decreases exocytosis ~40% but not the fraction of flash events which is shown in Figure 1F. Then the percentage of flash events should be >80% in Figure 1D for fsk+RP-8, but the flash percentage is similar as fsk alone with ~60-65%. How is this possible? This needs to be clarified to support the conclusion that Epac rather than PKA is responsible for cAMP-dependent fusion pore regulation.

We respectfully disagree. Assuming that PKA only affects the rate of exocytosis (but not fusion pore behavior), we would expect that the fraction of flash events remains constant in presence of the PKA blocker RP-8. This is in fact what is observed (Figure 1D).

At the end of Introduction, the authors write that activation of Epac2 via GLP-1 and sulfonylurea will hinder full fusion and that this is "expected to reduce their full fusion as insulin secretagogoues". However, there are no data available in this study (or in published elsewhere) showing that addition of GLP-1 or sulfonylurea will reduce insulin secretion. How do the authors validate the importance of their study? Is glucose stimulated insulin secretion decreased in the cells (Figure 2) overexpressing Epac2? What about Epac2 and insulin secretion in EndoC cells?

The statement is based on the data in Figure 5 suggesting that SU’s restrict fusion pore expansion by the same Epac-dependent mechanism as cAMP. In situations where exocytosis is strongly stimulated (in Figure 5 by SU, K^+^ and ex-4), we observe a relative decrease in exocytosis (and therefore likely also insulin release) rather than an additive effect of SU and fsk. In vivo, SU will primarily stimulate exocytosis by closing K_ATP_ channels and depolarization, but it will be worth investigating whether the stimulatory effect of GLP-1 agonists is affect by longterm SU treatment.

However, our study is focused on the molecular mechanism by which cAMP regulates fusion pore behavior, and we feel that in vivo studies to determine the physiological role of such mechanisms are beyond the scope of the paper. However, there is some evidence that fusion pore regulation by amisyn restricts insulin secretion, and may play a role in human type-2 diabetes (Collins, 2016). We also discuss the possibility that a delay between the release of small transmitters and insulin after Epac activation could be physiologically relevant, for example by influencing para/autocrine signaling in the islet. Such effects might be difficult to detect with traditional insulin secretion measurements, in particular since small molecule release and exocytosis are not considered in secretion measurements.

Regarding EndoC cells, please refer to our comment above.

The efficiency of the overexpression is not shown. What is the mRNA and protein levels of Epac2 (Figure 2), amysin and dynamin (Figure 6) after overexpression?

We believe that such quantification should be done on single cells, because the transfection only reaches a (varying) fraction of the cells and expression levels vary from cell to cell. We have therefore immunostained INS1 cells that had been transfected with mCherry-amisyn. In these cells we quantified both mCherry-amisyn and immunostaining (anti-amisyn detected with Alexa 488) in a large number of cells, including non-expressing cells (a). As expected, the Alexa fluorescence is higher in transfected cells (b), and there is a near linear relationship between red and green fluorescence (c); the y-axis offset is caused by non-transfected cells that are labeled green but not red, and represents endogenous amisyn expression. From this, we estimate that we worked with cells that overexpress the mCherry-amisyn 2-4 fold compared with endogenous expression. The data also hint that endogenous amisyn may be suppressed by expression of mCherry-amisyn (the average of non-transfected cells is higher than the offset). Similar experiments were done with dynamin1-GFP, and we observed a similar 2-3 fold increase for overexpressed dynamin 1 (d and e). The range of overexpression is similar to a number of other proteins that we have tested (eg syntaxin, CaV1.2), but unfortunately we were unable to source Epac antibodies of sufficient quality to succeed with such quantifications for Epac2.

The authors have not performed the proper statistics for the data. A simple t-test is not enough when comparing multiple conditions. At least one-way ANOVA with ad hoc-test for multiple comparison.

We thank the reviewer for the suggestion. We now provide one-way ANOVA with posthoc tests (Kruskal Wallis/Dunns) instead of t-tests. The new statistics are described in the methods and legends.

The authors used Epac2 KO mice established in Kopperud et al. 2017, in which the characteristics of microvascular permeability of these mice was examined. However, the characteristics regarding glucose homeostasis including insulin secretion were not well studied in the paper. Furthermore, other lines of Epac2 KO mice have already been established and analyzed. The authors need to examine at least the insulin secretory profiles in the Epac2 KO mice used in this study in vivo and in vitro, and discuss them by referring to the previous studies using other KO strains.

The Epac2 knockout (KO) mouse used in this study differs from the original and most widely employed strain in that the deletion involves exons 12-13, which encode the high-affinity cAMP binding domain present in all Epac2 isoforms. With the previously generated mouse, a deletion was made in the first exon, thereby eliminating the dominating longest isoform Epac2A, but allowing expression of the shorter isoforms Epac2B and 2C. Indeed, there is evidence that Epac2B is present in mouse pancreatic islets, although it has not been clarified in which cell type(s) the protein is expressed Høivik et al., 2013. We therefore think that it is advantageous to use the mouse in which all Epac2 isoforms have been eliminated. We have added a short description in the methods section explaining the difference between the present mouse and the Epac2A-KO mouse.

A general characterization of the Epac2-KO mouse is ongoing but beyond the scope of this manuscript. Available insulin secretion data do not indicate any striking phenotypical differences from the first published Epac2A-KO strain. We have performed intraperitoneal glucose tolerance tests showing that the mice are slightly glucose intolerant at an age of 10-11 months (Author response image 2). This is reminiscent of the glucose intolerance reported for Epac2A-KO following increased metabolic demand induced by e.g. high fat feeding (Song et al., 2013).

Moreover, our analyses show that glucose-stimulated insulin secretion from isolated islets remains intact (Author response image 3 and Author response image 4), whereas amplification of secretion by cAMP-elevating glucagon is impaired (Author response image 4). Although the number of observations are too few for the difference to reach statistical significance, the trend is as expected in light of the function of Epac2 and previously published data from Epac2A-deficient islets. We think that this functional characterization of the mouse would not add much to the present manuscript and think that the data would fit better in a forthcoming manuscript with more extensive phenotypic analyses and functional comparison between Epac2 and Epac1.

**Author response image 2. respfig2:** Means ± s.e.m. for blood glucose during intraperitoneal glucose tolerance tests in three 10-11-month-old wild type and Epac2-deficient mice.

**Author response image 3. respfig3:** Means ± s.e.m. for insulin from isolated islets stimulated by an elevation of the glucose concentration from 3 to 11 mM. N=4 mice for each genotype.

**Author response image 4. respfig4:** Means ± s.e.m. for blood for insulin from isolated islets stimulated by an elevation of the glucose concentration from 3 to 20 mM and addition of 10 nM glucagon. N=3 mice for each genotype.

Figure 2 shows that the percentage of flash events is increased by overexpression of Epac2. However, the actual number of flash events is not changed but full fusion events are decreased (Figure 2B), resulting in a decrease in the total number of fusion events (Figure 2A). In Figure 1, Epac activation by S223 increases the total number of exocytotic events by increasing flash events without changing the number of full fusion events. It seems that the fusion pore-regulating mechanism is different between these two conditions. Any thoughts on this issue?

We speculate that Epac becomes inhibitory for exocytosis due to the strong activation of pore restriction with both fsk and exogenous Epac, which competes with the cAMP-dependent acceleration of priming/exocytosis. We believe that this is also the reason for decreased exocytosis in presence of both forskolin and sulfonylurea (Figure 5B). Collins et al., (2016, and comment by Barg and Gucek, 2016) came to a similar conclusion regarding the role of amisyn in type 2 diabetes.

In Figure 4, the total number of exocytotic events is not increased by fsk in Epac2 KO β-cells. However, it is unlikely that cAMP cannot increase the number of exocytotic events in the absence of Epac2, as PKA signaling remains intact. What is the authors' interpretation of these results?

We are also surprised by the weak forskolin effect and the explanation is not clear. Since the PKA-dependent effect is slower than the Epac effect on exocytosis (Renström, 1997), it may be speculated that it has not become manifested during the relatively short time course of the experiment. Another possibility is that cAMP is already somewhat elevated in the control situation, since the cells are exposed to 10 mM glucose. Elevated cAMP may lead to PKA activation. It has been demonstrated that the apparent Kd for stimulating granule priming/exocytosis is 5-fold lower for PKA than for Epac (Eliasson, 2003).

In Figure 4F and 4G, fsk seems to prolong fusion pore lifetime significantly in both WT and Epac2 KO β-cells, although the authors state that the lifetimes are similar with or without forskolin. This should be modified.

Thank you for pointing this out. While the ratios of flash events are equally low, the NPY release time is increased in Epac2KO mice after addition of fsk. We have edited the text accordingly:

“In contrast, exocytosis was not accelerated by forskolin in Epac2^-/-^ cells, and the fraction of flash events was similar with or without forskolin (Figure 4C, F-G).” (subsection “cAMP-dependent fusion pore regulation is absent in Epac2^-/-^ (*Rapgef4*-/-) β-cells”).

In Figure 5, glib, tolb and gliz show similar effects on fusion pore regulation. However, the gliclazide data are inconsistent with the previous reports in which gliclazide among various sulfonylureas does not activate Epac2 (Zhang et al., 2009, Takahashi et al., 2013, Takahashi et al., 2013). The results should be confirmed using β-cells from Epac2 KO mice.

We´ve performed the requested experiment, where we bathed Epac2KO β-cells with gliclizide and monitored exocytosis upon K^+^ depolarization. We observed no increase in either fraction of flash events (a), exocytosis rate (b) or the NPY release times (c), suggesting that Epac2 is required for the effect of gliclazide on the fusion pore. We had already included similar measurements with tolbutamide in the Epac2 KO model (Figure 7 bottom left), indicating that both SU’s affect the pore via Epac2.

**Author response image 5. respfig5:** 

The authors show in Figure 6 that dynamin and amisyn are translocated to the site of exocytosis in a cAMP-dependent manner, and suggest that Epac protein is preassembled at the site of the secretory machinery. The authors also mention the involvement of Epac2 in the recruitment of dynamin and amisyn to the site of exocytosis, based on the observations of the translocation of these proteins by cAMP stimulation, which do not provide sufficient evidence to draw the authors' conclusion. The authors should show the behaviors of dynamin and amisyn in Epac2 KO β-cells and when Epac is activated pharmacologically.

Unfortunately, we are unable to perform the suggested experiments in Epac2KO cells, without first generating and testing an adenoviral expression system for labeled dynamin and amisyn. We have instead performed experiments in INS1-cells, in which we quantified amisyn or dynamin binding in presence of S223 or ESI-09 to pharmacologically modulate Epac function.

As evident from the new Figure 6, pharmacological inactivation with ESI09 prevents binding of either dynamin1 (c) or amisyn (f) to the release site, resulting in decreased NPY release times (d,g) and fractions of flash events (e,h). Similarly, in absence of fsk, pharmacological activation of Epac2 by S223 results in accumulation of proteins at the release site at the point of fusion (i,l) and longer NPY release times (j,m) together with higher fraction of flashes (k,n). We´ve changed the figure legends accordingly (changes are highlighted in yellow). The results strengthen the conclusion that Epac activation leads to accumulation of dynamin and amisyn at the release site.

The significance of the restriction of fusion pore expansion is not clear. Both full fusion and flash events eventually release all contents of the secretory vesicle. Thus, the prolonged fusion pore lifetime means only a few seconds delay in insulin release compared to release of small molecules (or small molecules release a few seconds in advance of the insulin release). Although the authors discuss the roles of these small molecules in autocrine and paracrine signaling in islets, the physiological and pathophysiological relevance of the time difference in the release of insulin and small molecules should be further discussed.

The Discussion section has been in part rewritten to strengthen the focus on physiological and pathophysiological relevance.

It has previously been shown that there are different modes of insulin exocytosis, based on the dynamics of insulin granules (predocked granules, newcomer granules, etc.) (Shibasaki et al.,2007 and Nagamatsu et al., 2006). Is there any difference in the mode of insulin exocytosis between the present study and the previous reports? The authors might want to discuss the modes of exocytotic events found in the present study in relation to previous studies.

While a number of labs have reported crash fusion (all using similar protocols), others have not (Tsuboi and Rutter, 2003; Barg et al., 2002; Michael et al., 2007; Hoppa et al., 2009; Gandasi and Barg, 2014). We recently presented a comprehensive analysis of near-membrane granule behavior in human β-cells, where there was no evidence for crash fusion (Gandasi et al., 2018). The reason why some groups observe crash fusion but others not is unclear. Possible differences include the use of glucose instead of KCl for stimulation (Shibasaki et al., 2007), the duration of the stimulation, and technical issues such as the use of thick cell supports (eg laminin), mistargeting of the granule label to other vesicle types, and label bleaching during long recordings. The latter would give the false appearance of no granule at the site of exocytosis. While this question is interesting and important (and should be settled), we believe that it is not central to the mechanism of fusion pore regulation by Epac and should rather be discussed in a topical review.

[Editors' note: further revisions were requested prior to acceptance, as described below.]

Thank you for resubmitting your work entitled "Fusion pore regulation by cAMP/Epac2 controls cargo release during insulin exocytosis" for further consideration at eLife. Your revised article has been favorably evaluated by Vivek Malhotra (Senior Editor), a Reviewing Editor, and three reviewers.The manuscript has been improved but there are some remaining issues that need to be addressed before acceptance, as outlined below:1) Figure 1B. The drawing of full fusion with a wide concave structure may be subject to error, as recent super-resolution STED imaging shows that fusing dense-core vesicles maintain an omega-shape till the STED resolution limit (Shin et al., 2018; Chiang et al., 2014). We suggest to re-draw the concave shape in Figure 1B as an omega shape with a larger pore than the other omega profile at the left side. This does not mean that full-collapse does not occur in β-cells, but the authors' data can only say that the fusion pore is larger than NPY-venus.

We thank the reviewer for the suggestion. We have redrawn the final fusion step in Figure 1B as requested.

2) The reviewer's comment: “Figure 2 shows that the percentage of flash events…”The authors do not respond to the comment properly. The reviewers asked the authors to discuss the mechanisms of the different findings between Figure 2 and Figure 1. This should be reconsidered.

This is probably due to its independent action on priming and fusion pore control. With activation (Figure 1; S223) both the effect of priming and the effect on the fusion pore are strongly present. With Epac overexpression only (Figure 2B), there is no effect on the priming, but only on the fusion pore restriction. As we´ve discussed in the response letter, we speculate that Epac, similarly as amisyn, becomes inhibitory for exocytosis due to the strong restriction of the pore with both fsk and exogenous Epac present. We might be under-detecting the restricted fusion pores that close back (kiss-and-run events), which accounts for the observed exocytosis decrease.

We´ve added a short discussion in the Discussion section:” Epac mediates the PKA-independent stimulation of exocytosis by cAMP ^59^ and our data suggests it may affect both priming and fusion pore restriction.”

3) The reviewer's comment: “In Figure 5, Glib, tolb and gliz show similar effects…”The authors have performed the requested experiments, and suggested that Epac2 is required for the effect of gliz on the fusion pore. The result on gliz is different from the previous reports (Zhang et al., 2009; Takahashi et al., 2013). The authors should discuss reasons for the discrepancy.

We thank the reviewer for the comment. The mentioned reports state gliz does not activate Epac2A, but it can still bind (only through 1 binding site G114) to the CNB1 domain (Takahashi et al., 2013). We´ve previously shown Epac2 targeting to the plasma membrane doesn’t depend on CNB1 domain. The clustering at the granule site however does depend on the CNB1 domain (Alenkvist et al., 2017). Since we speculate that Epac works only as a regulator of downstream proteins (e.g. dynamin 1 and amisyn), the localization at the granule site without activation may explain the restriction of the fusion pores we´ve observed in this study. This is further supported with the data in Figure 2, where overexpression of Epac in absence of forskolin results in a strong restriction of the fusion pores.

We´ve added a short discussion in the Discussion section: “Since gliclizide binds the CNB1 domain without activating it ^54^ and still restricts the fusion pore, Epac localization at the granule site ^39^ may be enough to regulate the downstream proteins (e.g. dynamin and amisyn).”

4) The reviewer's comment: “The role of Eoac2(cAMP-GEFII) in insulin exocytosis…” The authors responded to the comment, but the modified statement has not been incorporated in the text (Introduction).

We thank the reviewer for noticing the mistake. We´ve modified the text accordingly.

5) The reviewer's comment: “The source of reagents…” The source is provided, but the source of the Epac2 agonist S223 is missing.

We thank the reviewer for the comment. We´ve added the source of S223 (Biolog) to the key resource table.

6) The reviewer's comment: The description "inhibitors of the GLP-1 peptidase"… In the authors' response, "inhibitors of DPP-4 peptidase" should be "inhibitors of DPP-4"(delete"peptidase").

We´ve modified the text accordingly.